# SCoT: Teaching 3D-LLMs to Think Spatially with Million-Scale CoT Annotations

**Jinpeng Li[1]*  Haiping Wang[1,*]  Jiabin Chen[1]  Yuan Liu[2]  Zhen Dong[1,†]  Bisheng Yang[1,†]**
[1]LIESMARS, Wuhan University  [2]Hong Kong University of Science and Technology

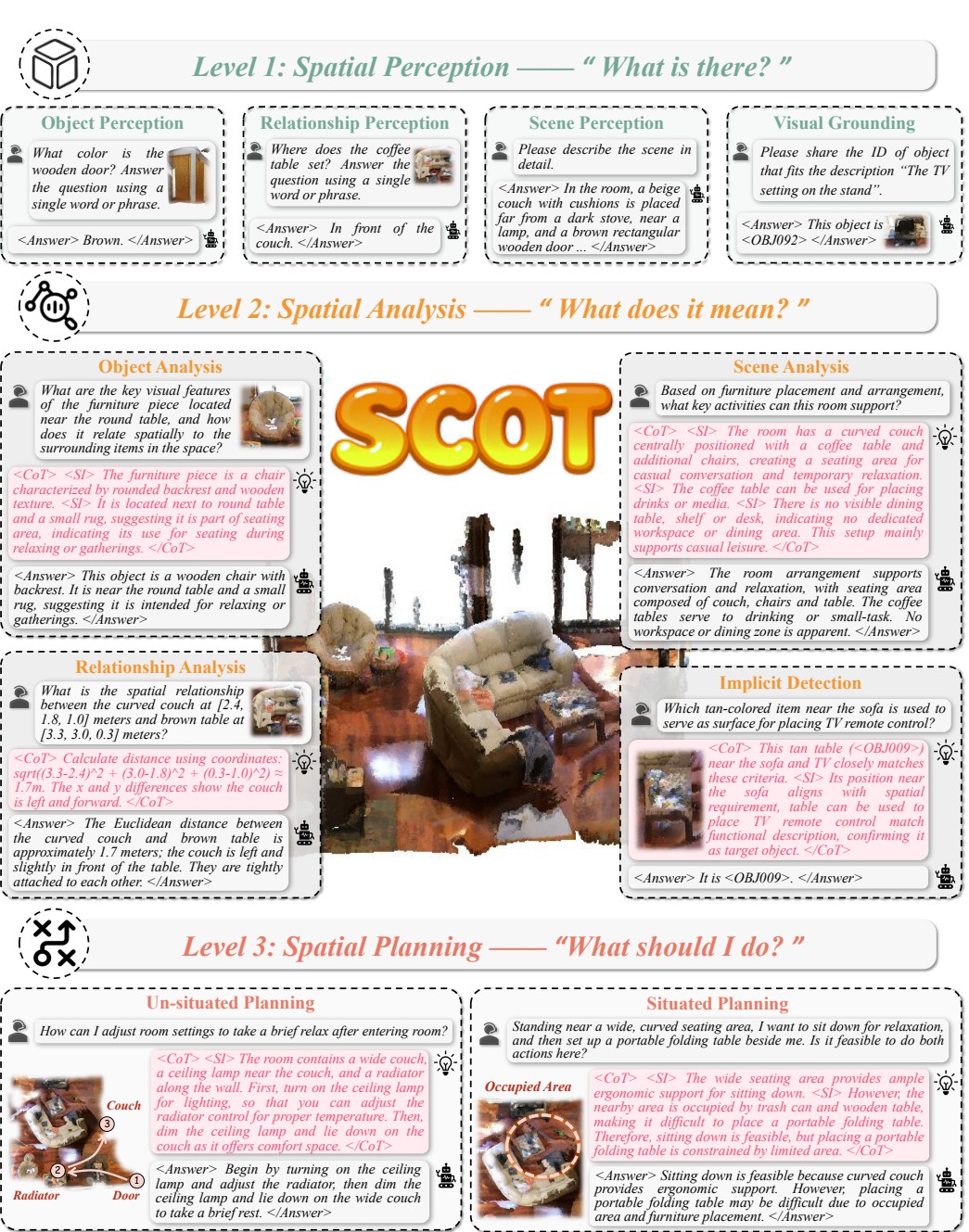

Figure 1: SCoT provides 1.1M cases to supervise 3D-LLMs in learning spatial perception, analysis, and planning. Beyond query–answer pairs, each analysis and planning case explicitly includes a Chain-of-Thought (CoT) annotation grounded in the scene context.

---

*The first two authors contribute equally.
†Corresponding Authors.

## ABSTRACT

Recent advances in 3D Large Language Models (3D-LLMs) show strong potential in understanding and interacting with 3D environments, yet their training data typically lack explicit reasoning processes, limiting complex spatial reasoning and task planning. To address this, we annotate SCoT, a million-scale Chain-of-Thought dataset spanning three levels: a) *Spatial Perception* (what is there), recognizing object properties, relations, and scene attributes; b) *Spatial Analysis* (what does it mean), inferring rationality, functionalities, and physical implications; c) *Spatial Planning* (what should I do), integrating perception and reasoning for actionable strategies. Unlike prior datasets supervising only answers, SCoT annotates intermediate reasoning grounded in scene cues, specifically for analysis and planning tasks. Results show that CoT supervision greatly benefits complex analysis and planning but induces hallucinations and accuracy drops in simple perception. These findings highlight both the necessity and the nuanced challenges of scene-grounded reasoning for advancing 3D intelligence. The dataset and code are available at `https://github.com/WHU-USI3DV/SCoT`.

## 1 INTRODUCTION

The burgeoning field of 3D Large Language Model (3D-LLM) learning aims to equip LLMs with a human-like understanding of and ability to interact with the three-dimensional world. While LLMs have revolutionized text and 2D image understanding, their extension into 3D realms remains superficial. Current 3D-LLMs are primarily trained on datasets that provide only question-answer (Q-A) pairs (Table 1), effectively teaching models to recall and associate but not to reason. This paradigm reduces powerful models to black boxes that output answers without transparency into their decision-making processes, severely limiting their reliability and applicability in real-world scenarios, such as robotics and embodied AI, where trust and verifiability are paramount.

Chain-of-Thought (CoT) reasoning is a powerful method for enhancing model transparency, achieving remarkable success in textual and 2D visual domains. However, its direct transfer to 3D understanding is non-trivial. Unsupervised CoT frequently produces rationales that are linguistically plausible but scenically ungrounded, relying on linguistic priors rather than actual 3D evidence. This results in a new type of black box: one that appears interpretable yet fails to remain faithful, ultimately undermining trust (Huang et al., 2025c).

Thus, some existing works annotate CoT chains to enforce scene-grounded CoT. Yet, these methods suffer from two key limitations. First, they do not differentiate between simple perceptual tasks and complex reasoning tasks. However, we empirically show that indiscriminately applying CoT to simple perceptual tasks—where direct observation suffices—can add unnecessary complexity and induce hallucinations (Huang et al., 2025b; Liu et al., 2025a; 2024b), yielding 4.9% accuracy drop as shown in Table 2 and Fig. 2. Second, their limited scale in Table 1, prevents models from learning diverse and robust reasoning patterns.

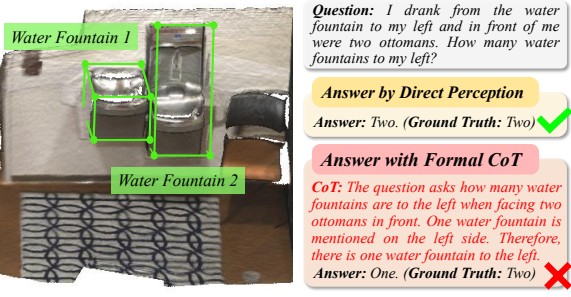

Figure 2: Overuse of CoT in simple perceptual tasks can introduce hallucinations, overlooks, and errors.

To address these issues, we first introduce a principled three-tier taxonomy that reflects a natural progression in task complexity and explicitly dictates when CoT should be employed—enhancing reasoning where needed while avoiding overcomplication where unnecessary:

• *Spatial Perception ("What is there?")*: This layer focuses on foundational scene understanding, including object properties (e.g., color, shape), object-to-object relations (e.g., left of, supporting), and global scene attributes (e.g., furniture composition). Since these tasks require few inference beyond direct observation, we supervise the model using answer-only training to strengthen its grounding in visual evidence, avoiding unnecessary CoT that could introduce hallucinations.

- *Spatial Analysis ("What does it mean?")*: At this level, the model must interpret perceived elements by integrating them with world knowledge. Tasks include inferring functionality, causality, physical stability, and spatial logic (e.g., "This is a dining room because the table is surrounded by chairs"). Here, we introduce CoT to explicitly guide the model through structured reasoning steps.

- *Spatial Planning ("What should I do?")*: The highest layer requires synthesizing perception and reasoning into actionable plans. The model must generate sequential strategies for embodied agents (e.g., "To clean this room, first pick up the clothes, then. . . "). For these tasks, CoT is essential to articulate the logical flow from observation to action.

Guided by this taxonomy, we construct SCoT (Spatial Chain-of-Thought), a *million-scale* dataset designed to teach 3D-LLMs how to think spatially, not just what to answer. *For analysis and planning*, SCoT asks CoT reasoning grounded on real scene information. To build it, given a scene, we first construct a scene context. Specifically, the scene is decomposed into objects, represented as a scene graph, and described with rich textual descriptions. Together with the BEV scene image, this scene context is used to prompt an VLM to generate "Question–CoT–Answer" samples. To explicitly demonstrate how CoT reasoning leverages scene information, we require the CoT to be grounded in the scene context and introduce a dedicated <SI> token. Whenever scene information (e.g., object properties, spatial relationships, structural attributes) is utilized, it must be cited with an <SI> tag. In this way, the CoT transforms vague assertions into verifiable, grounded claims. For instance, instead of stating "The table is too small", the model is expected to reason as follows: "The table <SI>(with a width of 0.8m)</SI> is too small to fit <SI>all six chairs (each 0.5m wide)</SI> around it."

We train the baselines and a self-designed model on this SCoT dataset. Experiments demonstrate that SCoT, evolving from answer-supervised to grounded-CoT training, achieves significant performance gains across benchmarks. More importantly, it yields models whose reasoning processes are transparent, faithful to the scene, and fundamentally more explainable and trustworthy.

## 2 RELATED WORK

**3D Large Language Models.** LLMs show strong reasoning and dialogue abilities, and extending them to 3D perception has become a key paradigm for natural language interaction in 3D environments. Early point-cloud LLMs (Xu et al., 2024; Guo et al., 2023; Wang et al., 2023) align point clouds with text to support object-level queries, but struggle with scene-level reasoning. Hybrid 3D Vision-Language models (Zhu et al., 2024; Deng et al., 2025) combine multi-view image features from 2D backbones (e.g., LLaVA (Liu et al., 2023)) with 3D position-aware point cloud features, while Video LLMs (Huang et al., 2024; Zheng et al., 2025) incorporate 3D spatial cues into video representations to capture semantics and relations, enabling more complex spatial reasoning.

**Trainsets for 3D-LLM.** Advances in 3D-LLM have driven the creation of 3D-Language datasets linking 3D perception with natural language. Early works introduced task-specific benchmarks for captioning (Chen et al., 2021; Jia et al., 2024), question answering (Azuma et al., 2022; Ma et al., 2022; Zhao et al., 2022; Linghu et al., 2024), and grounding (Chen et al., 2020; Zhang et al., 2023; Achlioptas et al., 2020). However, these resources restrict cross-task transfer in 3D LLMs. Recent efforts have shifted toward large-scale multi-task datasets (Li et al., 2023; Wang et al., 2024; Lyu et al., 2024; Yang et al., 2025a; Huang et al., 2025a; Zhang et al., 2025; Gong et al., 2025), supporting dense captioning, multi-turn dialogue, and embodied planning. 3D-CoT (Chen et al., 2025) further emphasizes reasoning by adding CoT annotations for object-level tasks, yet remains limited to object-centric understanding. Importantly, most datasets still lack explicit reasoning traces or grounding in 3D evidence, leaving scene-level CoT reasoning largely unexplored.

**Chain-of-Thought Reasoning.** CoT prompting (Kojima et al., 2022; Wei et al., 2022) has proven effective in eliciting step-by-step reasoning for text tasks. Reinforcement learning further strengthens CoT reliability, as shown in OpenAI o1 (Jaech et al., 2024) and DeepSeek-R1 (Guo et al., 2025). Inspired by these successes, recent work extends CoT to vision-language models (Yang et al., 2025b; Huang et al., 2025d; Shen et al., 2025; Peng et al., 2025; Liu et al., 2025b). In 3D-LLM, CoT reasoning remains nascent but promising: approaches such as 3D-R1 (Huang et al., 2025c), SpaceR (Ouyang et al., 2025), and Spatial-MLLM (Wu et al., 2025) augment datasets with CoT annotation or apply reinforcement learning with task-specific rewards, achieving chain-based outputs. Yet, current studies are constrained by dataset scale, task diversity, and transparency, highlighting the need for broader, high-quality 3D CoT resources covering perception, reasoning, and planning.

Table 1: Comparison of SCoT dataset and the existing 3D large language model train sets. "M.3D." means the Matterport3D dataset, "3RS." means the 3RScan dataset, "M.S." means the MultiScan dataset, "S.3D." means the Structured3D dataset, "P.T." means the ProcTHOR dataset, "Obj." means the Objaverse dataset, "ARK." means the ARKitScenes dataset, "3D.F." means the 3D-Front dataset. "Reasoning" refers to the reasoning process besides the final answer.

| Dataset | Source | Scene | Size | Reasoning | Tasks |
|---|---|---|---|---|---|
| ScanRefer (ECCV 2020) | ScanNet | 800 | 32k | | Grounding |
| Scan2Cap (CVPR 2021) | ScanNet | 800 | 32k | | Caption |
| ScanQA (CVPR 2022) | ScanNet | 800 | 27k | | QA |
| SQA3D (ICLR 2023) | ScanNet | 800 | 33k | | QA |
| 3D-LLM (NeurIPS 2023) | ScanNet | 1513 | 659k | | Multiple Tasks |
| EmbodiedScan (CVPR 2024) | ScanNet, M.3D. | 5185 | 1M | | Multiple Tasks |
| M3Dbench (ECCV 2024) | ScanNet, M.3D. | 700 | 327k | | Multiple Tasks |
| SceneVerse (ECCV 2024) | ScanNet, 3RS., M.S., S.3D., P.T. | 97000 | 2.4M | | Multiple Tasks |
| LEO (ICML 2024) | ScanNet, 3RS., M.S., Obj. | 3000 | 513k | | Multiple Tasks |
| MSR3D (NeurIPS 2024) | ScanNet, 3RS., ARK. | 1700 | 251k | | Multiple Tasks |
| 3D-GRAND (CVPR 2025) | ScanNet, S.3D., 3D.F. | 40000 | 6.2M | | Multiple Tasks |
| 3D-CoT (Arxiv) | - | - | 1.6M | ✓ | Object QA |
| 3D-R1 (Arxiv) | ScanNet, 3RS., M.S., S.3D., P.T. | 1513 | 30k | ✓ | Multiple Tasks |
| SpaceR-151k (Arxiv) | ScanNet, General Image and Video | - | 151k | ✓ | Multiple Tasks |
| Spatial-MLLM-120k (NeurIPS 2025) | ScanNet, ARK. | - | 120k | ✓ | Multiple Tasks |
| SCoT (ours) | ScanNet | 800 | 1.1M | ✓ | Multiple Tasks |

## 3  SCoT CONSTRUCTION

With approximately 1.1 million samples, SCoT provides grounded factual descriptions and reasoning chains spanning 3D intelligent tasks. We first introduce our three-tier taxonomy of 3D task classification to analyze their core characteristics and suitability for CoT reasoning. Based on this taxonomy, we construct task-specific "Query–CoT–Answer" data and introduce our quality control procedures.

### 3.1  TASK TAXONOMY

**Spatial Perception** (240K samples) tells "*What is there*". It only requires basic scene observation. We reveal that excessive use of CoT introduces hallucinations in perception tasks, as shown in Sec. 5.2. Therefore, by default, we discard CoT and adopt answer-only supervision for perception. We provide CoT-annotated perception data to validate this observation empirically. The perception tasks include: (a) *Object Perception:* Identifying explicit properties of individual objects (color, shape, size, spatial location); (b) *Relationship Perception:* Recognizing spatial and semantic relationships between objects (adjacency, containment, support); (c) *Scene Perception:* Understanding holistic scene attributes (room type, furniture composition, global arrangements); (d) *Visual Grounding:* Localizing objects based on text descriptions explicitly mentioning the object.

**Spatial Analysis** (460K samples) tells "*What does it mean*". It requires inference beyond direct observation, incorporating logical and quantitative reasoning chains grounded in 3D contexts, including: (a) *Object Analysis:* Conducting in-depth analysis of specific objects, including their attributes beyond visualization and potential functions; (b) *Relationship Analysis:* Quantitatively assessing relationships between object pairs by calculation (e.g., relative distances, orientations, topological dependencies); (c) *Scene Analysis:* Human-like reasoning at the scene level, such as layout evaluation, capacity analysis, space assessment, or improvement suggestions; (d) *Implicit Detection:* Localizing objects based on implicit text descriptions without explicit name or semantic mentions.

**Spatial Planning** (390K samples) tells "*What should I do*". It involves action sequencing and task execution planning within realistic environmental constraints, including: (a) *Un-situated Planning:* Planning and decision-making under additional requirements (e.g., power outages, blockages, water supply interruptions) without giving the situated conditions, requiring to decompose complex objectives into manageable sequential steps; (b) *Situated Planning:* Assessing the feasibility and procedure for executing a specified task, given the agent's current state (including its position, orientation, and ongoing activities), allowing the use of spatial terms and interaction-based analysis.

This structured taxonomy enables comprehensive coverage of 3D reasoning capabilities, from basic perception to complex planning. Most tasks in existing 3D vision-language benchmarks can be mapped to one or more of these categories. By providing a unified framework for task definition and evaluation, the SCoT dataset facilitates systematic and interpretable implementation of 3D LLMs.

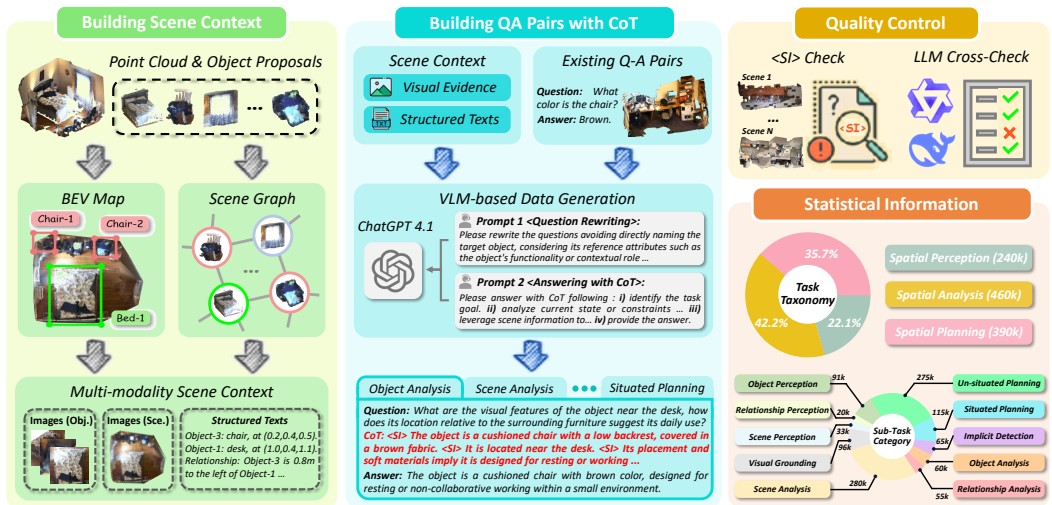

Figure 3: Annotation and data statistics of SCoT. SCoT first constructs a detailed scene context for each 3D scene as input to the VLM, then prompts the VLM to generate Query–CoT–Answer cases grounded in the scene. In total, we built 1.1M samples spanning perception, analysis, and planning.

## 3.2 DATA GENERATION

As discussed above, for *Spatial Perception* tasks, we collect and adapt existing Q-A pairs from publicly available 3D-VL benchmarks, specifically ScanRefer, ScanQA, SQA3D and Scan2Cap. For *Spatial Reasoning* and *Planning* tasks, we leverage advanced VLMs to generate Q-A pairs along with CoT reasoning chains. The CoT generation process begins by providing the VLM with rich descriptions of 3D scene context, including object location, spatial relationships, and global layout information. Using this context, the VLM is guided to generate "Query–CoT–Answer" samples that incorporate multi-step reasoning and in-depth scene analysis.

**Building Scene Context.** Given a 3D scene, we first segmented it into object proposals. Each object is assigned a bounding box and a semantic label. These objects are organized into a scene graph, where nodes represent objects and edges encode spatial relations such as adjacency, support, and relative direction. These attributes are serialized into textual templates (e.g., "Object-3 [chair, at (0.2, 0.4, 0.5)] is 0.8m to the left of Object-1 [desk, at (1.0, 0.4, 1.1)]"). In parallel, the scene is projected into a 2D Bird's-Eye View (BEV) map, where each object is highlighted with a unique identifier and bounding box. Local object-centric crops and global scene images further supply visual evidence. Finally, all elements above are consolidated and expressed as structured texts paired with visual evidence, serving as our multi-modality scene context with rich scene information.

**Building QA Pairs with CoT.** Existing datasets typically provide only simple answer supervision, leaving the thinking process as a black box. To construct explicit reasoning traces, we leverage the powerful capabilities and knowledge of VLMs. Specifically, we provide the VLM with the scene context as a reference, together with a simple Q-A pair for expansion by the following two steps:

● *Question Rewriting*: In existing datasets, queries often explicitly mention target objects by name, which allows the LLM to directly locate the object and makes the reasoning process trivial. To fully stimulate 3D-LLM's scene perception and reasoning ability, we need to prepare queries avoiding directly naming the target. Instead, the queries should reference attributes such as the object's functionality or contextual role. For example, "What color is the chair?" is reformulated as "What are the visual features of the object near the desk, how does its location relative to the surrounding furniture suggest its daily use?" In tasks related to spatial planning, VLM is further prompted to infer object states or simulate real-world conditions.

● *Answering with CoT*: The VLM needs to provide step-by-step reasoning chains to reach the answer, transforming concise responses into comprehensive logical explanations. To maintain quality and consistency, we prompt the VLM to answer with: *"i) identify the task goal. ii) analyze the current state or constraints and extract the object and scene content. iii) leverage scene information to form the basis for spatial analysis or planning. iv) finally, provide the answer."*

**Asking CoT Explicitly Grounded on Scene Context.** Naive CoT has a risk of hallucination and superficial reasoning, where generated rationales may appear logically consistent but are detached

Table 2: Results on *SCoT-Perception* tasks with or without perception CoT supervision. "CoT" means using CoT annotations in the perception tasks. The model is SCoT-Reasoner.

| CoT | ScanRefer | | Multi3DRefer | | Scan2Cap | | ScanQA | | | SQA3D |
|---|---|---|---|---|---|---|---|---|---|---|
| | Acc@0.25 | Acc@0.50 | F1@0.25 | F1@0.50 | CIDEr@0.5 | B-4@0.5 | CIDEr | B-4 | EM | EM |
| | **58.4** | **53.4** | **60.1** | **55.7** | **79.6** | **38.2** | **87.9** | **15.0** | **23.0** | **55.8** |
| ✓ | 56.4 | 51.3 | 53.5 | 49.3 | 75.7 | 35.8 | 73.4 | 13.1 | 21.9 | 47.4 |

Table 3: Overall performance comparison on *SCoT-Analysis* tasks requiring text output. "R.", "M.", "Exp.", "Fai." and "Tru." are abbreviations for "ROUGE-L", "METEOR", "Explainability", "Faithfulness" and "Trustworthiness", respectively. Note that the score range for "Explainability", "Faithfulness", and "Trustworthiness" metrics is from 1 to 10. "(CoT)" means the method trained with "Full SCoT Setting", and the counterpart without it is trained with "Answer-Only Setting".

| Method | Object Analysis | | | | | Relationship Analysis | | | | | Scene Analysis | | | | |
|---|---|---|---|---|---|---|---|---|---|---|---|---|---|---|---|
| | R. | M. | Exp. | Fai. | Tru. | R. | M. | Exp. | Fai. | Tru. | R. | M. | Exp. | Fai. | Tru. |
| Chat3D V2 | 24.91 | 15.42 | 6.71 | 5.15 | 6.08 | 26.03 | 13.48 | 4.71 | 3.45 | 4.10 | 20.38 | 13.16 | 6.73 | 6.48 | 6.52 |
| Chat3D V2 (CoT) | 25.18 | 15.76 | 6.93 | 5.50 | 6.35 | 26.23 | 14.91 | 4.87 | 4.07 | 4.54 | 21.55 | 13.97 | 6.74 | 6.43 | 6.58 |
| Video 3D LLM | 19.72 | 12.93 | 6.84 | 5.02 | 5.97 | 25.71 | 14.07 | 4.42 | 3.03 | 3.98 | 22.01 | 12.79 | 7.28 | 7.01 | 6.92 |
| Video 3D LLM (CoT) | 20.57 | 14.07 | 7.02 | 5.43 | 6.24 | 26.90 | 14.92 | 4.74 | 3.11 | 4.02 | 22.85 | 13.06 | 7.72 | 7.33 | 7.37 |
| Chat Scene | 26.11 | 15.57 | 6.35 | 5.32 | 5.63 | 27.17 | 15.63 | 4.98 | 4.32 | 4.45 | 21.84 | 14.04 | 7.71 | 7.20 | 7.23 |
| Chat Scene (CoT) | 26.98 | 15.80 | 6.75 | 5.80 | 6.07 | 28.40 | 16.15 | 5.62 | 5.29 | 5.30 | 22.72 | 15.10 | 7.80 | 7.51 | 7.49 |
| SCoT-Reasoner | **27.34** | 15.62 | 6.67 | 5.59 | 5.89 | 29.71 | 16.41 | 5.19 | 3.77 | 4.23 | 22.59 | 14.68 | 7.82 | 7.32 | 7.45 |
| SCoT-Reasoner (CoT) | 27.22 | **16.17** | **7.04** | **6.15** | **6.41** | **30.69** | **16.60** | **5.77** | **5.34** | **5.41** | **23.48** | **15.29** | **7.95** | **7.55** | **7.68** |

from actual scene content, even scene contexts were provided. We thus introduce a <SI> (Scene Information) tagging mechanism in CoT generation process, the VLM is strictly prompted to explicitly insert <SI> at every step in the reasoning chain where scene-derived information (e.g., object properties, spatial relations or scene layout) is utilized, as illustrated in Fig. 1.

On the one hand, this approach enforces grounded reasoning, requiring the reasoning process to remain faithful to the visual context; samples with insufficient or incorrect use of <SI> tags are discarded. On the other hand, it enhances traceability, enabling users and developers to verify which elements of the scene underpin each inference.

## 3.3 QUALITY CONTROL

Besides the above <SI> check, to improve reliability of SCoT, we perform LLM-based cross-check. We first employ ChatGPT-4.1 (Achiam et al., 2023), Qwen (Bai et al., 2023) and DeepSeek (Liu et al., 2024a) as independent agents. During cross-check process, if any of the agents identify issues (e.g., ill-posed question, incorrect or misleading CoT, ambiguous answer, or inconsistencies between the scene facts and answer or CoT), these samples are flagged and removed. This strict procedure ensures that the SCoT dataset maintains the highest standards of data quality and factual consistency.

Afterwards, we conduct manual accuracy evaluation. We randomly select 50 samples for each task among trainset, resulting in a total of 500 samples to undergo manual checking. After excluding problematic cases, 447 samples are retained, yielding an acceptance rate of approximately 90%.

## 4 TRAINING 3D-LLMS WITH SCoT

**3D-LLM Selection**. To more comprehensively validate SCoT, we train and evaluate various 3D-LLM architectures on SCoT, including 3D VG Transformer (Zhao et al., 2021), Chat 3D V2 (Huang et al., 2023), Chat Scene (Huang et al., 2024), and Video 3D-LLM (Zheng et al., 2025). Introductions to these baselines are provided in the Appendix (Sec.A.8). Moreover, we find that most prior methods focus on a single spatial modality (e.g., point clouds or images). To more comprehensively evaluate the effectiveness of SCoT, we additionally design SCoT-Reasoner, a unified framework supporting multimodal inputs such as images and point clouds. Its architecture and implementation details are provided in the Appendix (Sec.A.2).

**Training Strategy**. We adopt a two-stage training strategy to effectively integrate structured reasoning into 3D-LLMs. In the first stage, the model is trained on spatial perception samples from SCoT to establish a robust understanding of object properties, relationships, and scene structure. In the second stage, the model is fine-tuned on spatial analysis and planning tasks, where it learns to generate explicit, step-by-step reasoning chains grounded in scene information.

Table 4: Overall performance comparison on *SCoT-Planning*. "R.", "M.", "Exp.", "Fai." and "Tru." are abbreviations for "ROUGE-L", "METEOR", "Explainability", "Faithfulness" and "Trustworthiness", respectively. Note that the score range for "Explainability", "Faithfulness", and "Trustworthiness" metrics is from 1 to 10. "(CoT)" means the method trained with "Full SCoT Setting", and the counterpart without it is trained with "Answer-Only Setting".

| Method | Situated Planning | | | | | Un-situated Planning | | | | |
|---|---|---|---|---|---|---|---|---|---|---|
| | R. | M. | Exp. | Fai. | Tru. | R. | M. | Exp. | Fai. | Tru. |
| Chat3D V2 | 23.93 | 12.21 | 6.38 | 6.18 | 6.33 | 28.69 | 16.44 | 6.08 | 5.88 | 6.03 |
| Chat3D V2 (CoT) | 24.95 | 13.01 | 6.65 | 6.34 | 6.56 | 28.99 | 16.23 | 6.35 | 6.04 | 6.26 |
| Video 3D LLM | 21.66 | 11.79 | 6.02 | 5.87 | 5.91 | 26.82 | 14.29 | 6.20 | 6.07 | 5.74 |
| Video 3D LLM (CoT) | 21.34 | 12.35 | 6.31 | 6.10 | 6.00 | 27.29 | 15.90 | 7.01 | 6.45 | 6.19 |
| Chat Scene | 24.87 | 12.92 | 6.47 | 6.49 | 6.35 | 27.73 | 16.09 | 6.80 | 6.94 | 6.72 |
| Chat Scene (CoT) | 25.02 | **13.26** | 7.02 | **7.05** | 6.95 | 27.24 | 16.16 | 6.96 | 7.08 | 6.91 |
| SCoT-Reasoner | 24.21 | 12.37 | 6.64 | 6.09 | 6.30 | 28.01 | **18.35** | 6.97 | 6.93 | 6.76 |
| SCoT-Reasoner (CoT) | **25.13** | 13.06 | **7.38** | 6.94 | **7.14** | **29.04** | 18.11 | **7.27** | **7.29** | **7.15** |

## 5 EXPERIMENTS

### 5.1 EXPERIMENTAL PROTOCOLS

**Settings.** To ensure a comprehensive evaluation of SCoT, each model in Section 4 is re-trained in two distinct configurations: (a) *Full SCoT Setting*: Models are trained using the complete SCoT data, including both answers and structured CoT sequences. (b) *Answer-Only Setting*: Models are trained using only the final answers from SCoT, with CoT sequences removed, to ablate and evaluate the specific contribution of CoT supervision. All the experiments are implemented with PyTorch on a single NVIDIA A100 GPU. Training process of the first and second stage takes about 6 and 28 hours to converge, respectively. All other settings are guaranteed the same for a fair comparison.

**Datasets.** To facilitate a comprehensive evaluation of model capabilities across different cognitive levels, we split the SCoT test set into three dedicated subsets: (a) *SCoT-Perception* is curated from existing public datasets, incorporating simple Q-A pairs from ScanQA, SQA3D, ScanRefer, Multi3DRefer, and Scan2Cap. It focuses on tasks such as visual question answering, caption generation, and explicit visual grounding, serving to evaluate basic scene perception and understanding. (b) *SCoT-Analysis* aims to assess model capabilities in object-centric, relationship-oriented, and scene-level reasoning, as well as implicit detection. This test set contains only final answers without chain-of-thought sequences, enabling pure outcome-based evaluation. (c) *SCoT-Planning* is designed to evaluate un-situated and situated planning abilities. Similarly, it provides only final answers without intermediate reasoning chains, focusing the assessment on plan correctness and feasibility.

**Metrics.** For *Traditional Metrics*, we report Acc@0.25 and Acc@0.50 for grounding tasks. For text-based tasks, we adopt metrics including BLEU, ROUGE-L, METEOR, EM, and CIDEr. However, these metrics are proven unstable for only capturing text similarities (Qi et al., 2025; Lovin, 2025).

Thus, we further adopt more reliable *Comprehensive Metrics*. Motivated by recent works (Qi et al., 2025; Lovin, 2025), we introduce three comprehensive criteria evaluated by LLMs: (a) *Explainability*: Measures the clarity and logical coherence of the response, ensuring the reasoning process is transparent and easy to follow; (b) *Faithfulness*: Evaluates whether the response remains strictly grounded in the provided scene evidence without introducing unsupported claims or hallucinations; (c) *Trustworthiness*: Assesses the overall credibility and reliability of the answer, considering both factual accuracy and presentation style. We prompt ChatGPT, Qwen, and DeepSeek as evaluators. Each model receives the generated response, ground truth, and relevant scene context. Each criterion is rated on a scale of $1-10$, with final scores averaged across all evaluators. For MSQA, we report the average correctness score across test sets following Linghu et al. (2024).

### 5.2 QUANTITATIVE RESULTS

*SCoT-Perception*. Consistent with analysis in Sec.3.1, supervising CoT construction for simple perception tasks leads to 4.9% performance drop, as shown in Table 2. In these cases, linguistic priors may override scene-grounded cues, while the forced multi-step reasoning introduces unnecessary steps that further increase hallucinations. More results on SCoT-Perception with multiple baselines are provided in the Appendix (Sec.A.5).

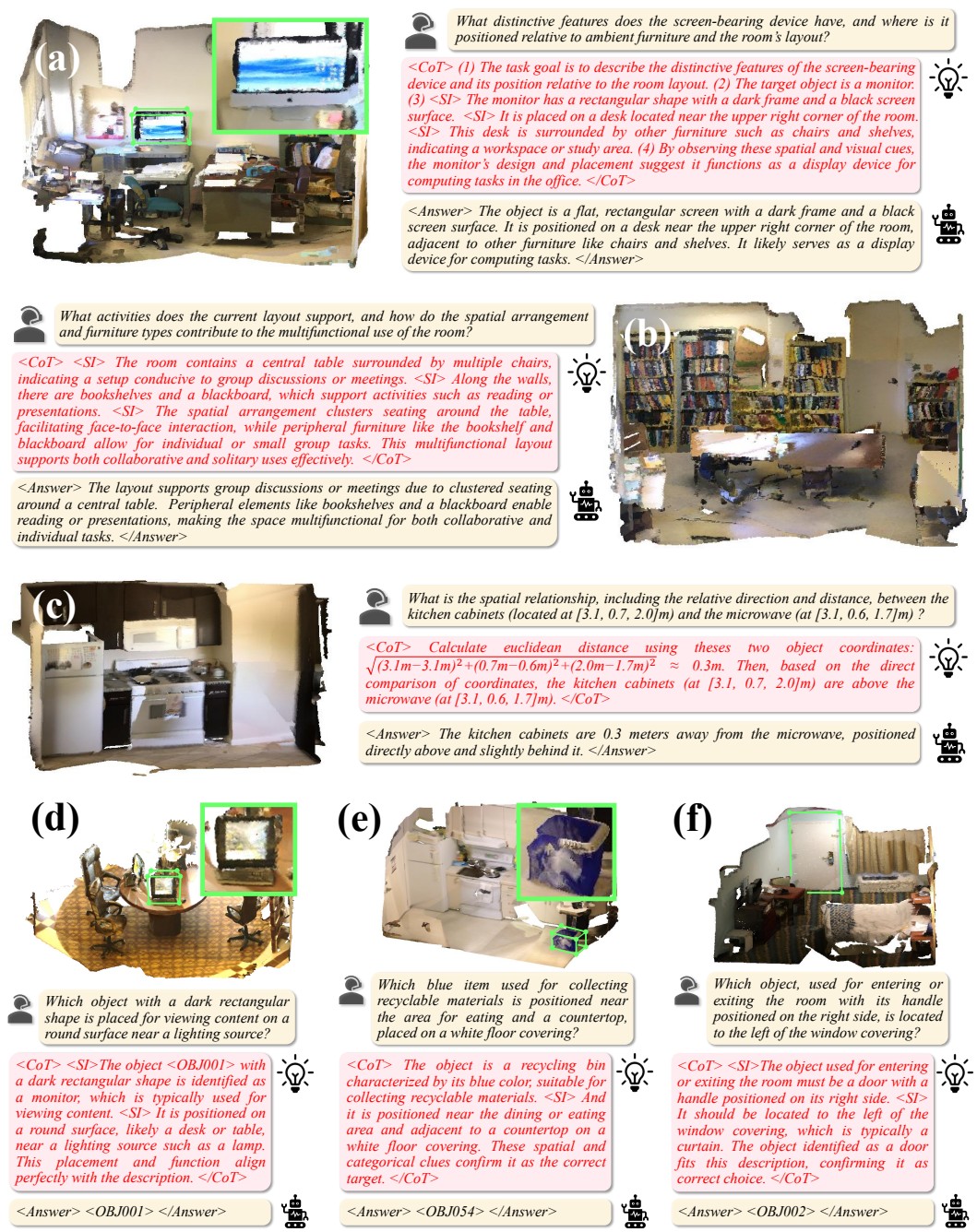

Figure 4: Representative qualitative results on spatial analysis tasks with CoT setting. (a) object analysis, (b) scene analysis, (c) relationship analysis, (d-f) implicit detection.

*SCoT-Analysis and SCoT-Planning.* Table 3 and Table 4 summarize the performances of baselines trained with or without CoT supervision. Consistent with conclusions obtained in previous studies (Wei et al., 2022), models with smaller parameter scales, such as 7B, exhibit minimal differences in held-in-text similarity metrics. For instance, the SCoT-Reasoner with CoT setting yields only modest enhancement over the non-CoT setting in text similarity metrics, with average improvement of 0.74% in ROUGE-L and 0.36% in METEOR. However, training with CoT consistently leads to substantial gains in comprehensive scores, with average increases of 6.21%, 11.74%, and 10.02% in explainability, faithfulness and trustworthiness, indicating the response answers generated after CoT reasoning process align better with human-like cognition and achieve higher accuracy.

We further observe that CoT plays an indispensable role in the implicit detection task, as it enables the model to infer the target object in implicit queries with statements that do not explicitly mention object

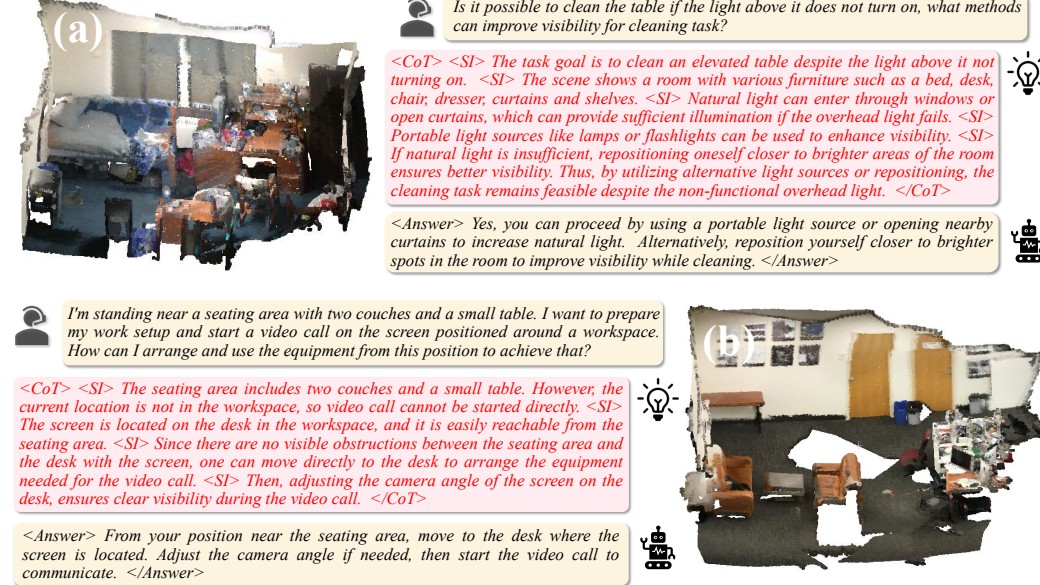

Figure 5: Results on planning tasks with CoT setting. (a) un-situated planning, (b) situated planning.

names before outputting the result. As shown in Table 5, the CoT setting significantly outperforms the non-CoT setting, with improvements of 23.6% and 20.7% of Acc@0.25 and Acc@0.50 metrics for SCoT-Reasoner, respectively.

## 5.3 QUALITATIVE RESULTS

Table 5: Quantitative results on implicit detection task in *SCoT-Analysis*.

| Method | Implicit Detection | |
|---|---|---|
| | Acc@0.25 | Acc@0.50 |
| 3D VG Transformer | 3.0 | 2.7 |
| 3D VG Transformer (CoT) | 12.5 | 10.9 |
| Chat Scene | 6.7 | 5.3 |
| Chat Scene (CoT) | 26.9 | 23.2 |
| Video 3D LLM | 6.1 | 4.9 |
| Video 3D LLM (CoT) | 24.1 | 20.9 |
| SCoT-Reasoner | 8.6 | 7.7 |
| SCoT-Reasoner (CoT) | **32.2** | **28.4** |

Figure 3 presents qualitative results of SCoT-Reasoner on the *SCoT-Analysis* test set, with additional cases provided in the appendix. For *object analysis* in Fig. 4 (a), the model begins by interpreting task objective in CoT, then extracts reliable scene information, including object appearance, spatial position, and nearby objects, and uses this evidence to infer the object's potential function. For *scene analysis* in Fig. 4 (b), its CoT thoroughly analyzes furniture composition and spatial layout in CoT to infer the room's functional type. For *relationship analysis*, the reasoning can also estimate inter-object distances and relative orientations directly from numerical coordinates, as shown in Fig. 4 (c). Fig. 4 (d-f) presents qualitative results for *implicit detection*. When the query does not explicitly specify the object name, the CoT can infer its category, evaluates its alignment with the description across multiple attributes, and ultimately identify the correct target object.

For *un-situated planning* shown in Fig. 5 (a), when the light fails to function properly during the cleaning task, the reasoning observes a curtained window and suggests opening it for natural light or using a portable power source. For *situated planning* shown in Fig. 5 (b), CoT generates a multi-step plan starting from the current location mentioned in the prompt to accomplish a complex goal.

## 5.4 GENERALIZABLE REASONING FROM SCoT

We further verify whether SCoT has truly learned transferable scene reasoning patterns rather than merely memorizing ScanNet scenes or language templates. Based on the MSQA benchmark (Linghu et al., 2024), we evaluate its generalizability from three perspectives.

Table 6: Evaluation results on the MSQA test set under ScanNet and ARKitScenes environments. ‡ indicates zero-shot test. Methods annotated with "(·)" indicate the 3D-VL dataset used for model training (e.g., "(SCoT)" or "(3D-R1)"), then is evaluated on MSQA to assess its generalization across different 3D environments.

| Method | Counting | Existence | Attributes | Spatial | Navigation | Others | Overall |
|---|---|---|---|---|---|---|---|
| *MSQA-ScanNet* | | | | | | | |
| (a) LEO | 32.5 | 88.5 | 58.7 | 44.2 | 39.6 | 81.4 | 54.8 |
| (b) MSR3D | 32.3 | **93.1** | 50.0 | 46.5 | 54.1 | 75.6 | 54.2 |
| (c) LEO-VL | 39.3 | 92.7 | 56.9 | **59.3** | **59.7** | 82.8 | **61.7** |
| (d) GPT-4o ‡ | 32.3 | 79.3 | **79.0** | 37.0 | 31.7 | **91.6** | 52.3 |
| (e) Chat Scene ‡ (3D-R1) | 35.3 | 41.9 | 34.4 | 45.1 | 25.2 | 86.2 | 43.1 |
| (f) Chat Scene ‡ (SCoT) | 35.5 | 64.8 | 44.5 | 42.7 | 39.8 | 84.8 | 47.6 |
| (g) SCoT-Reasoner ‡ (SCoT) | **41.5** | 63.5 | 54.8 | 48.9 | 43.0 | 91.2 | 54.4 |
| *MSQA-ARKitScenes* | | | | | | | |
| (h) GPT-4o ‡ | 37.5 | **55.2** | 48.1 | 37.7 | 21.0 | 60.7 | 41.0 |
| (i) Qwen-VL ‡ | **43.2** | 46.0 | 44.5 | 25.3 | 26.9 | **70.1** | 39.7 |
| (j) SCoT-Reasoner ‡ (SCoT) | 31.2 | 47.5 | **52.6** | **40.5** | **29.5** | 65.9 | **41.2** |

**Q-A Format Pattern Generalization (ScanNet → MSQA–ScanNet)**. Although the underlying scenes remain the same, the question formats, instruction modes, and description structures of MSQA–ScanNet differ completely from SCoT. The results in Table 6 indicate that SCoT-Reasoner (g), trained on SCoT dataset, achieves a score of 54.4, which is comparable to in-domain supervised models like LEO (a) and MSR3D (b). Furthermore, it surpasses the zero-shot performance of large-scale GPT-4o (d) by 2.1 points. These findings demonstrate that the inference process does not rely on the language templates or annotation frameworks used during training; instead, it suggests that SCoT has learned genuine scene logic and reasoning structures rather than mere textual patterns.

**Cross-Scene Generalization (ScanNet → ARKitScenes)**. Scenes in MSQA–ARKitScenes dataset have a completely different environment, sensor characteristics, and geometric quality from ScanNet. As shown in Table 6, SCoT-Reasoner (j) trained with SCoT achieves a total score of 41.2 on MSQA–ARKitScenes, outperforming the zero-shot capabilities of larger-scale LVLMs such as GPT-4o (h) and Qwen-VL (i). Notably, it secures a +3.3 improvement in reasoning-intensive categories like Attributes, Spatial, and Navigation. Qualitative results in Fig. 10 of Appendix further reveal that the model can execute stable multi-step spatial reasoning even amidst major changes in geometric structure and viewpoint. This indicates that the supervision provided by SCoT facilitates the learning of spatial reasoning mechanisms that are transferable across domains, rather than structural memorization of ScanNet.

**Compared to Other Reasoning Datasets (SCoT vs. 3D-R1)**. To eliminate differences in models and training strategies, we trained a third-party Chat Scene model (Huang et al., 2024) using 3D-R1 (Huang et al., 2025c) and SCoT under strictly consistent configurations and performed zero-shot evaluation on the out-of-domain MSQA-ScanNet dataset (as shown in Table 6 (e) and (f)). The results show that SCoT significantly outperforms 3D-R1, particularly in categories requiring complex reasoning, such as Navigation (+14.6) that needs multi-step state tracking, spatial relationship understanding, and grounded inference. This evidence proves that SCoT provides higher-quality and more generalizable reasoning supervision.

# 6 CONCLUSION

We introduced SCoT, a million-scale Chain-of-Thought dataset annotated to supervise 3D-LLMs with structured, scene-grounded reasoning chain. SCoT covers diverse tasks spanning perception, analysis, and planning. Our experiments demonstrate substantial gains over counterparts trained without CoT annotation and supervision, especially for analysis and planning, highlighting the value of explicit reasoning supervision for advancing complex spatial intelligence in 3D environments. Our future work will involve improving the inference efficiency in real-time applications through model compression (e.g., quantization and lightweight distillation), hardware-aware optimization, and task-adaptive reasoning modes.

## 7    ETHICS STATEMENT

This work does not involve human subjects, personally identifiable information, or sensitive data. The datasets used are publicly available 3D-LLM benchmarks and self-constructed SCoT, and all data collection and processing strictly comply with their respective licenses. No potential conflicts of interest or ethical concerns were identified in relation to this study.

## 8    REPRODUCIBILITY STATEMENT

Our work consists of a dataset SCoT and a model SCoT-Reasoner. To ensure reproducibility, we provide in the appendix detailed descriptions of the dataset construction pipeline, including data generation procedures and prompts, which enable others to replicate the dataset. The model design details are also comprehensively documented in the appendix. Furthermore, we plan to release the dataset and related resources publicly to facilitate reproducibility and future research.

## 9    ACKNOWLEDGMENTS

This research was jointly supported by the National Natural Science Foundation of China (General Program) (Grant No. 42571521), the NSFC Basic Research Project for Ph.D. Candidate (Grant No. 424B2012) and the Wuhan Natural Science Foundation Project (2025041001010363).

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

# A    APPENDIX

In the appendix, we provide more statistical information about SCoT dataset, a detailed introduction of SCoT-Reasoner, and quantitative results in *SCoT-Perception* dataset. Moreover, we present additional visualization results to further demonstrate the effectiveness of the CoT setting.

## A.1    SCoT DATASET

### A.1.1    STATISTICAL INFORMATION OF SCoT DATASET

Table 7: Statistical overview of SCoT dataset. We report the number of samples, average question length (Que. Len.), average CoT length (CoT Len.), and average answer length (Ans. Len.) across different task categories. Note that in these tasks, the answer for implicit grounding task is only an object placeholder, hence the average word length of the answer is always one.

| Task Category | Samples | Words | | | Tokens | | |
|---|---|---|---|---|---|---|---|
| | | *Que. Len.* | *CoT Len.* | *Ans. Len.* | *Que. Len.* | *CoT Len.* | *Ans. Len.* |
| Spatial Perception* | 270k | 22.7 | 0.0 | 15.7 | 28.7 | 0.0 | 28.1 |
| Implicit Detection | 65k | 35.0 | 51.4 | 1.0 | 39.8 | 59.4 | 7.0 |
| Object Analysis | 60k | 27.1 | 85.4 | 45.5 | 29.9 | 105.4 | 53.3 |
| Relationship Analysis | 55k | 25.7 | 32.1 | 27.8 | 48.1 | 92.1 | 42.0 |
| Scene Analysis | 280k | 16.8 | 82.6 | 36.0 | 19.2 | 105.9 | 44.1 |
| Un-situated Planning | 210k | 22.3 | 92.0 | 31.6 | 24.9 | 114.9 | 37.1 |
| Situated Planning | 180k | 37.7 | 86.1 | 33.6 | 42.4 | 105.7 | 38.9 |

As shown in Table 7, we present the detailed statistics of SCoT dataset. Spatial perception tasks are primarily derived from simple Question–Answer datasets (e.g., ScanRefer (Chen et al., 2020), ScanQA (Azuma et al., 2022), SQA3D (Ma et al., 2022), Scan2Cap (Chen et al., 2021), and Multi3DRefer (Zhang et al., 2023)), which are designed to establish basic spatial understanding for 3D LLMs. However, these spatial perception tasks feature short responses, with an average of 15.7 words and 28.1 tokens, and contain no explicit CoT reasoning. In contrast, spatial analysis (object analysis, relationship analysis, scene analysis) and spatial planning (un-situated planning and situated planning) tasks with text-based output produce substantially longer answer responses. The shortest responses in these categories average 27.8 words and 37.1 tokens, supported by explicit CoT annotations that provide abundant scene-grounded evidence and transparent reasoning processes.

### A.1.2    PROMPT TEMPLATE FOR DATA GENERATION

As shown in Fig. 6 and Fig. 7, the prompt templates are guided through a carefully structured prompt that integrates task instructions, additional spatial information, and multi-view images. The system prompt defines the overall role of LLM, requiring it to produce a natural language question, a coherent reasoning chain, and a precise answer in a strictly formatted manner. The task description emphasizes the need to reason over spatial relationships, such as Euclidean distance and relative orientation, while ensuring clarity and logical progression in the output. Additional structured information, including object identifiers, coordinates, and bounding box dimensions, is incorporated to provide the model with explicit geometric cues for accurate reasoning. Finally, this enriched textual context is paired with multi-view images, ensuring that the model not only grounds its reasoning in visual evidence but also delivers results that are both interpretable and verifiable. This unified prompting strategy enables reliable 3D spatial understanding and consistent generation of detailed CoT explanations.

## A.2    SCoT-REASONER FOR MULTI-MODAL SPATIAL INPUTS

In this paper, we introduce a new 3D-LLM architecture named SCoT-Reasoner to process multi-modal inputs (e.g., text, point cloud, and video) and support spatial perception, analysis, and planning. We incorporate the "Object-Relationship-Scene" refinement module into SCoT-Reasoner to enhance the model's ability for spatial understanding.

**system_prompt:**
You are an advanced AI spatial reasoning system capable of generating object level question, detailed answers and reliable Chain-of-Thought (CoT).
**content_prompt:** # This needs to be changed depending on the different tasks.
[1] You must complete three steps:
Step 1:
(i) Generate a multifaceted question that ask about the target object, but without using object class names, semantic categories, or ID numbers.
(ii) Generate a question that ask the spatial relationship (including distance and relative direction) between two objects, using object name and location.
......
Step 2: Generate a precise answer for the corresponding question, may include:
(i) The object name and its appearance (ii) A plausible inference of object's function (iii) The space property of object (iv) Euclidean distance between the two objects and their relative spatial orientation (e.g., "Object A is 1.8 meters to the left and slightly in front of Object B") ...
Step 3: Generate a reliable CoT using object name, and detailed calculation or reasoning analysis process should be emphasized in CoT.
i) identify the task goal. ii) analyze current state or constraints and extract the object and scene content. iii) leverage scene information to form the basis for spatial analysis or planning. iv) finally, provide the answer. Note that whenever scene information (e.g., object properties, spatial relationships, structural attributes) is utilized, it must be cited with an <SI> tag preceding the corresponding sentence.
[2] You must follow these rules:
The question and answer should not focus on phrases like "This image..." or "Aerial view of..." or "The object highlighted by the red box...". Avoid using terms like 'highlighted object' in the question and answer. Do not include explanations or meta-commentary about the task itself. Finally, the output question needs to be preceded by a corresponding identifier mark <Question> and the end of the question needs to have an </Question> identifier mark, the output answer needs to be preceded by a corresponding identifier mark <Answer> and the end of the answer needs to have an </Answer> identifier mark. The output CoT needs to be preceded by a corresponding identifier mark <CoT> and the end of the CoT needs to have an </CoT> identifier mark. Please be careful not to have words such as 'top-down view' or 'BEV image'. Note that keep the length of answer to 20 to 30 words. Note that keep the length of CoT to 30 to 50 words. Please note that the question cannot refer to object class names, semantic categories, or ID numbers.
**additional_information_prompt:** # This needs to be changed depending on the different tasks.
The 'Object ID' and corresponding 'Name' of the target object highlighted by green box in <MultiView_Image>, and the 'Object ID' and corresponding 'Name' of the neighborhood object highlighted by red box in <MultiView_Image>.
The center point coordinates of the target object are " + str(<instance_coordinates>) + "m. The length, width and height of bounding box are " + str(<instance_boundingbox>) + "m. The span of the target object is " + str(max(<instance_boundingbox>)) + "m. The height of the target object is " + str(<instance_height>) + "m.
The center point coordinates of the neighborhood object are " + str(<nei_instance_coordinates>) + "m. The length, width and height of bounding box are " + str(<nei_instance_boundingbox>) + "m. The span of the neighborhood object is " + str(max(< nei_instance_boundingbox>)) + "m. The height of the neighborhood object is " + str(< nei_instance_height>) + "m.
**messages:**
"role": "system"  "content": **system_prompt**
"role":"user"  "content": **content_prompt + additional_information_prompt**
"type":'image_url'  "image_url": {'url':image/jpeg;base64, {base64_MultiView_Image}"}

Figure 6: Prompt templates for data generation on object analysis and relationship analysis tasks. Note that annotation statements are represented in green font, and non-textual (multi-modality) variables are represented in blue font.

**system_prompt:**
You are an advanced AI system capable of generating complex reasoning question, answer and reliable Chain-of-Thought (CoT) based on the 3D spatial configuration. Please generate question and answer pairs (e.g., Scene layout rationality, Scene spatial capacity, Scene functionality and Scene redesign suggestion, Scene planning, ...) that perform scene-level reasoning and planning.
**content_prompt:** # This needs to be changed depending on the different tasks.
[1] You must complete three steps:
Step 1: Generate the questions that involve higher-order reasoning about the entire scene layout, such as:
(1) Scene layout rationality: Is the current furniture or object arrangement practical or efficient?
(2) Scene spatial capacity: How much usable or unused space remains?
(3) Scene functionality: What functions (e.g., dining, sleeping, working) does the scene currently support?
(4) Scene redesign suggestion: What changes could be proposed to improve the scene?
(5) Scene planning: Based on the scene context and state (e.g., standing, sitting, holding an object, walking), accomplish a high-level task, such as exercising, cooking, cleaning, relaxing, or working.
......
Step 2: Generate precise answers, and the answers must be detailed, reasoned, and based on spatial arrangements, scene utilization, and functionality. The answer may include information such as object density, spatial zones used, remaining usable space, and suggestions for improvement.
Step 3: Generate a reliable CoT using object name, and detailed calculation or reasoning analysis process should be emphasized in CoT.
i) identify the task goal. ii) analyze current state or constraints and extract the object and scene content. iii) leverage scene information to form the basis for spatial analysis or planning. iv) finally, provide the answer. Note that whenever scene information (e.g., object properties, spatial relationships, structural attributes) is utilized, it must be cited with an <SI> tag preceding the corresponding sentence.
[2] You must follow these rules:
The question and answer should not focus on phrases like "This image..." or "Aerial view of..." or "The object highlighted by the red box...". Avoid using terms like 'highlighted object' in the question and answer. Do not include explanations or meta-commentary about the task itself. Finally, the output question needs to be preceded by a corresponding identifier mark (e.g., <Question: Scene layout rationality>) and the end of the question needs to have an identifier mark (e.g., </Question: Scene layout rationality>), the output answer needs to be preceded by a corresponding identifier mark (e.g., <Answer: Scene layout rationality>) and the end of the answer needs to have an identifier mark (e.g., </Answer: Scene layout rationality). Note that keep the length of answer to 40 to 50 words. Please note that the question cannot refer to object class names, semantic categories, or ID numbers.
**additional_information_prompt:** # This needs to be changed depending on the different tasks.
There are <object_1>, <object_2>,..., <object_N> in this scenario. The 'Object ID' and corresponding 'Name' of the target object highlighted by green box in <MultiView_Image>, and the 'Object ID' and corresponding 'Name' of the neighborhood object highlighted by red box in <MultiView_Image>.
In addition, there is an interrogative statement about the task in the original dataset (e.g., SQA3D, 3D-LLM dataset) that can be used as a reference for information extraction. The query is: <Question_Reference>, The answer is: <Answer_Reference>.
**messages:**
"role": "system"  "content": **system_prompt**
"role":"user"  "content": **content_prompt + additional_information_prompt**
"type":'image_url'  "image_url": {'url':image/jpeg;base64, {base64_MultiView_Image}"}

Figure 7: Prompt templates for data generation on scene analysis and spatial planning tasks. Note that annotation statements are represented in green font, and non-textual (multi-modality) variables are represented in blue font.

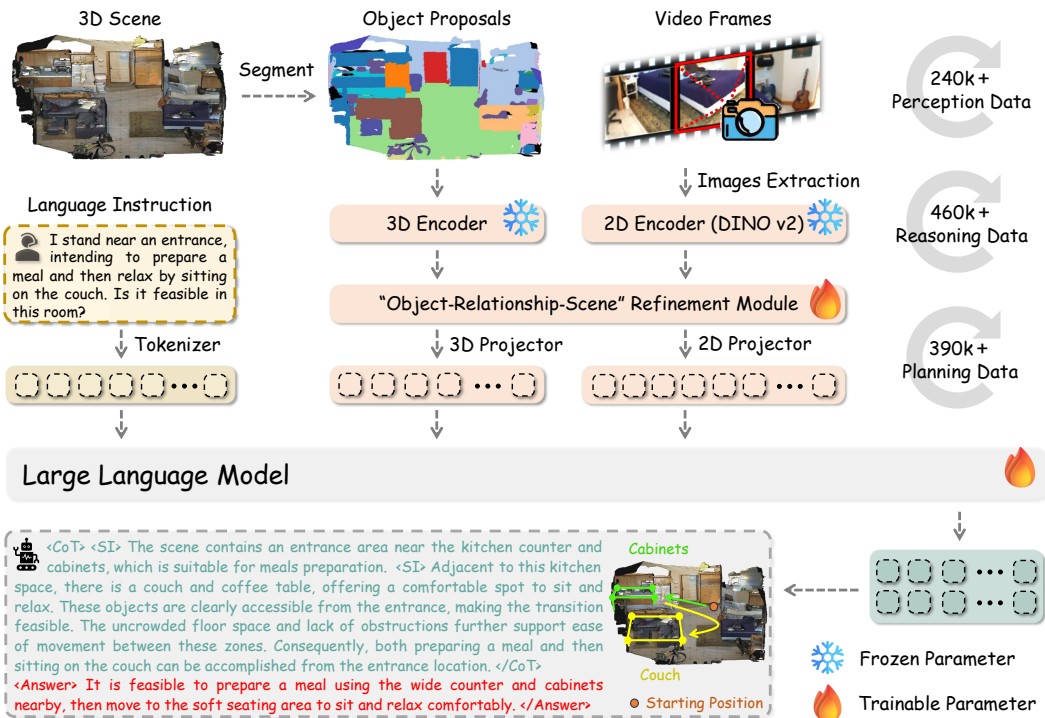

Figure 8: Model architecture of SCoT-Reasoner. SCoT-Reasoner receives language instruction, object proposals segmented from 3D scene and video frames as input, then performs step-by-step reasoning and analysis based on the object-grounded or scene-grounded facts, and ultimately generates reliable accurate and verifiable answers.

- **Text Tokenizer.** We employ Vicuna-7B to process text token sequences. An embedding look-up table is used to map these tokens into text embeddings.

- **3D Encoding.** To extract geometric and spatial attributes from 3D point clouds, we utilize Mask3D (Schult et al., 2022) to obtain 3D object proposals. For each object $P_i$, Uni3D (Zhou et al., 2023) is used to derive object-centric 3D feature vector $F_i^{3D} \in \mathbb{R}^{1 \times 1024}$.

- **Video Encoding.** For input video frame, we adopt open-vocabulary video segmentation model DEVA (Cheng et al., 2023) to extract 2D object proposals. For each detected region, we compute object-centric 2D feature vector $F_i^{2D} \in \mathbb{R}^{1 \times 1024}$ using DINOv2 (Oquab et al., 2023).

- **"Object-Relationship-Scene" Refinement Module.** We design the "object-relationship-scene" (ORS) refinement module that integrates absolute spatial locations and pairwise relational cues.

***Spatial Graph Construction.*** Each 3D object proposal is encoded by Uni3D into $F_i^{3D} \in \mathbb{R}^{1 \times 1024}$, while each tracked 2D object from video frames is encoded by DINOv2 into $F_i^{2D} \in \mathbb{R}^{1 \times 1024}$. We retain each object's 3D bounding box $B_i \in \mathbb{R}^{1 \times 6}$. We extract the object-center coordinates in $\mathbb{R}^{1 \times 3}$ and compute their pairwise differences to obtain a 3D offset vector, which is processed by a linear layer to generate a 3D offset embedding in $\mathbb{R}^{1 \times 1024}$. In the spatial graph, each object is represented as a node, while the offset embeddings constitute the edges.

***ORS Representation Optimization.*** Given the spatial graph constructed above, we apply Graph-Transformer to refine object-centric representations. The attention mechanism is augmented with a learnable spatial bias derived from offset embeddings, enabling each object to selectively aggregate information from spatially relevant neighbors, while preserving direction and distance cues. The spatially-refined feature $\hat{F}i \in \mathbb{R}^{1 \times 1024}$ obtained from the GraphTransformer is concatenated with the original $F_i^{2D}$ or $F_i^{3D}$ to produce an enhanced representation $\hat{F}_i^{ORS} \in \mathbb{R}^{1 \times 2048}$. The set of enhanced features $\{F_i^{ORS}\}_{i=1}^{N}$ forms the "object-relationship-scene" representation, which is subsequently projected into the LLM embedding space via modality-specific projectors.

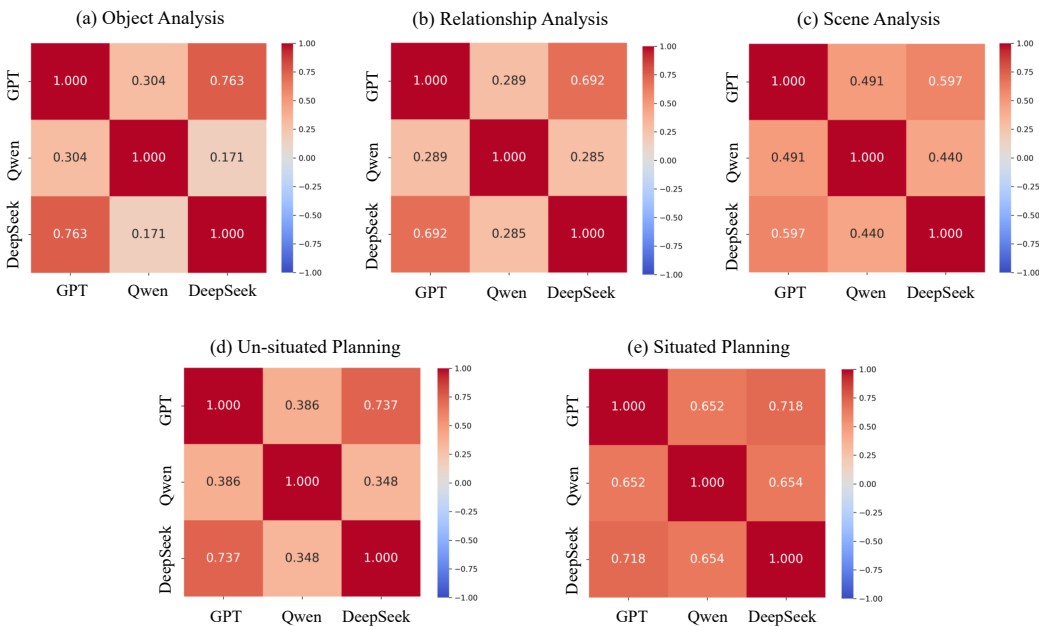

Figure 9: The correlation heatmap of LLM scores derived from ChatGPT-4.1, Qwen, DeepSeek across object analysis, relationship analysis, scene analysis, un-situated planning and situated planning tasks.

## A.3    IMPLEMENTATION DETAILS

We build SCoT-Reasoner upon the pretrained Vicuna-7B (v1.5) backbone, with additional projection layers for multimodal inputs. We add scene and image tokens to the input sequence and enable object identifiers, and the maximum number of objects is capped at 200. All fine-tuning process targeting the attention projections and feed-forward components is conducted with the LoRA layers (Hu et al., 2022). The LoRA rank is set to 64 with an $\alpha$ value of 16 and a dropout rate of 0.05. Both the textual embedding layer and LoRA parameters are trainable, while the image projection module remains frozen. For optimization, we use the AdamW optimizer with a learning rate of $5 \times 10^{-3}$, $\beta_1 = 0.9$, $\beta_2 = 0.999$, and weight decay of 0.02. A warm-up phase is applied for the first 10% of training epochs.

## A.4    INTER-EVALUATOR CORRELATION ANALYSIS

As illustrated in Fig. 9, we present the correlation heatmap visualization across the three heterogeneous evaluators (ChatGPT-4.1, Qwen, and DeepSeek), demonstrating that their rating remain independent. Despite receiving the same information (answer, ground truth, and structured scene evidences) and rating prompts as inputs, the scores derived from three evaluators show moderate but not absolute inter-evaluator correlation with the most values in 0.3 to 0.7, indicating that no shared model-specific preference dominates their evaluation process. By averaging the three scores, evaluator-specific bias are further reduced, yielding a more stable and objective assessment.

## A.5    QUANTITATIVE RESULTS IN SCoT-PERCEPTION DATASET

As shown in Table 8, we report the performances of SCoT-Reasoner and other baseline method on the *SCoT-Perception* test set. Note that here we do not apply the SCoT-style reasoning; we only evaluate and demonstrate the baseline perceptual abilities of different models.

SCoT-Reasoner achieves the best accuracy across both single-object and multi-object grounding benchmarks. It reaches an Acc@0.50 of 53.4% in the ScanRefer dataset (Chen et al., 2020), while it achieves an F1@0.50 of 55.7% in the Multi3DRefer dataset (Zhang et al., 2023). In addition, we further evaluate the methods on simple QA tasks such as ScanQA (Azuma et al., 2022),

Table 8: Quantitative results on spatial perception tasks. "SCoT-R" is the abbreviations for "SCoT-Reasoner". "(CoT)" means the method trained with "Full SCoT Setting", and "(w.o. CoT)" means the method trained with "Answer-Only Setting" without CoT reasoning.

| Method | ScanRefer | | Multi3DRefer | | Scan2Cap | | ScanQA | | | SQA3D |
|---|---|---|---|---|---|---|---|---|---|---|
| | Acc@0.25 | Acc@0.50 | F1@0.25 | F1@0.50 | CIDEr@0.5 | B-4@0.5 | CIDEr | B-4 | EM | EM |
| 3D-LLM | 30.3 | - | - | - | - | - | 59.2 | 7.2 | 20.4 | - |
| Chat 3D | - | - | - | - | - | - | 53.2 | 6.4 | - | - |
| Chat 3D v2 | 42.5 | 38.4 | 45.1 | 41.6 | 63.9 | 31.8 | 87.6 | 14.0 | 21.2 | 54.7 |
| LL3DA | - | - | - | - | 62.9 | 36.0 | 76.8 | 13.5 | - | - |
| Scene-LLM | - | - | - | - | - | - | 80.0 | 11.7 | 25.6 | 53.6 |
| Chat Scene | 55.5 | 50.2 | 57.1 | 52.4 | 77.1 | 36.3 | 87.7 | 14.3 | 21.6 | 54.6 |
| Video 3D LLM | 58.1 | 51.7 | 58.0 | 52.7 | **83.8** | **41.3** | **102.1** | **16.1** | **30.1** | **58.6** |
| SCoT-Reasoner | **58.4** | **53.4** | **60.1** | **55.7** | 79.6 | 38.2 | 87.9 | 15.0 | 23.0 | 55.8 |

Table 9: Ablation experiments on different CoT settings and <SI> levels. "R.", "M.", "Exp.", "Fai." and "Tru." are abbreviations for "ROUGE-L", "METEOR", "Explainability", "Faithfulness" and "Trustworthiness", respectively. "CoT.Len." means the average token length of CoT, and "<SI>" means average number of <SI> identifier in CoT. Relationship analysis is excluded from this comparison because its CoT mainly consists of numerical computation process.

| Task Category | R. | M. | Exp. | Fai. | Tru. | <SI> | CoT.Len. | Time (s) |
|---|---|---|---|---|---|---|---|---|
| Object Analysis (No CoT) | **27.34** | 15.62 | 6.67 | 5.59 | 5.89 | – | – | 6.51 |
| Object Analysis (w.o. Sce. in CoT) | 26.93 | 16.04 | 6.92 | 5.94 | 6.27 | 1.83 | 64.99 | 10.97 |
| Object Analysis (w.o. Obj. in CoT) | 26.85 | 15.34 | 6.43 | 5.37 | 5.72 | 1.91 | 85.36 | 11.71 |
| Object Analysis (Full CoT) | 27.22 | **16.17** | **7.04** | **6.15** | **6.41** | 2.83 | 122.45 | 15.98 |
| Scene Analysis (No CoT) | 22.59 | 14.68 | 7.82 | 7.32 | 7.45 | – | – | 5.08 |
| Scene Analysis (w.o. Sce. in CoT) | 22.50 | 14.37 | 7.67 | 7.40 | 7.37 | 2.45 | 66.98 | 10.26 |
| Scene Analysis (w.o. Obj. in CoT) | 22.91 | 14.50 | 7.91 | 7.35 | 7.60 | 1.97 | 51.55 | 9.15 |
| Scene Analysis (Full CoT) | **23.48** | **15.29** | **7.95** | **7.55** | **7.68** | 3.42 | 95.57 | 13.30 |
| Situated Planning (No CoT) | 24.21 | 12.37 | 6.64 | 6.09 | 6.30 | – | – | 6.42 |
| Situated Planning (w.o. Sce. in CoT) | 24.04 | 11.96 | 6.70 | 6.21 | 6.55 | 2.56 | 75.89 | 12.53 |
| Situated Planning (w.o. Obj. in CoT) | **25.37** | 12.54 | 7.10 | 6.85 | 7.07 | 1.99 | 49.67 | 10.90 |
| Situated Planning (Full CoT) | 25.13 | **13.06** | **7.38** | **6.94** | **7.14** | 3.56 | 99.89 | 13.95 |
| Un-situated Planning (No CoT) | 28.01 | 18.35 | 6.97 | 6.93 | 6.76 | – | – | 6.15 |
| Un-situated Planning (w.o. Sce. in CoT) | 28.34 | 17.89 | 6.82 | 7.17 | 6.98 | 2.90 | 96.36 | 12.81 |
| Un-situated Planning (w.o. Obj. in CoT) | 28.85 | **18.40** | 7.20 | 7.05 | **7.31** | 1.82 | 52.31 | 9.57 |
| Un-situated Planning (Full CoT) | **29.04** | 18.11 | **7.27** | **7.29** | 7.15 | 3.91 | 120.28 | 14.27 |

Scan2Cap (Chen et al., 2021), and SQA3D (Ma et al., 2022), where the answers are relatively short. On these datasets, SCoT-Reasoner achieves competitive performance, ranking slightly below Video 3D LLM (Zheng et al., 2025), which benefits from temporal cues and richer video-based training.

## A.6 ABLATION STUDY FOR DIFFERENT COT SETTINGS AND <SI> LEVELS

As shown in Table 9, to have a deep understanding of SCoT reasoning process, we explore the effects of object-level reasoning and scene-level reasoning process. Specifically, we categorized, filtered, and ablated different types of <SI> annotations. "w/o Obj" denotes removing <SI> for object-centric information, while "w/o Sce" removes scene-level analysis <SI> from the CoT. The quantitative results indicate:

• Object-level reasoning is critical for object-centric tasks; removing it consistently reduces performance (e.g., METEOR 16.17→15.34; Explainability 7.04→6.43; Faithfulness 6.15→5.37).

• Scene-level reasoning is essential for holistic scene understanding and planning, with substantial drops when removed (e.g., Scene Analysis METEOR 15.29→14.37; Situated Planning Trustworthiness 7.14→6.55).

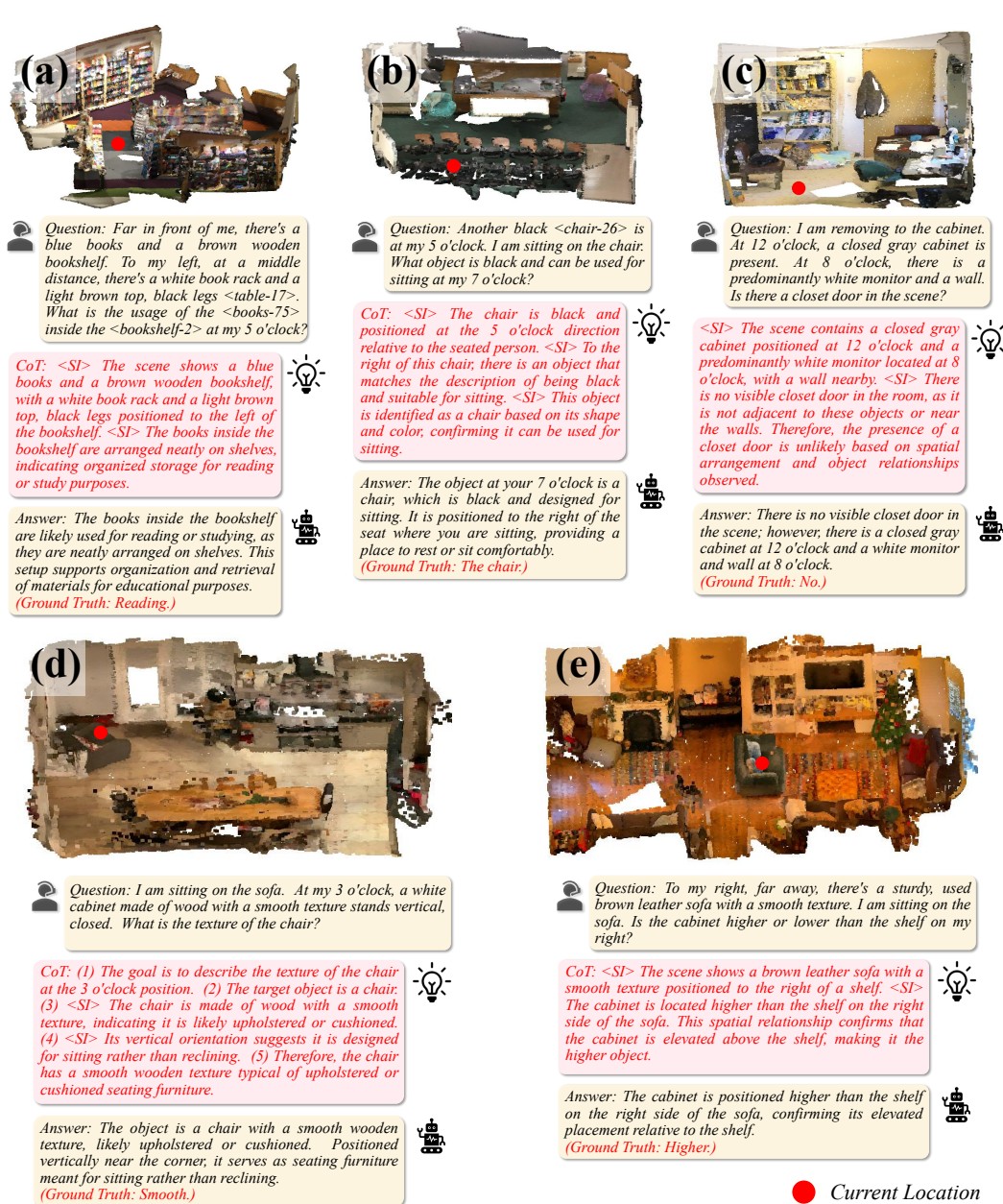

Figure 10: Qualitative generalization results of SCoT-Reasoner on different scenes in MSR3D (MSQA) dataset. (a-c): ScanNet, (d) and (e): ARKitScenes. Note that SCoT-Reasoner is trained on SCoT dataset.

However, the metric improvements from CoT reasoning comes with higher inference latency. Inference time increases by 2.0×–3.2×, depending on task complexity. For example, the inference time of scene analysis increases from 5.08s to 13.30s, and un-situated planning increases from 6.15s to 14.27s on a single A100 GPU. These tasks inherently require deeper reasoning, making the additional overhead acceptable for offline planning but challenging for time-critical robotics applications.

## A.7 ADDITIONAL QUALITATIVE RESULTS

As illustrated in Fig. 11, the additional qualitative results of object analysis indicate how CoT enhances the interpretability and reliability of responses in reasoning-oriented tasks. Rather than providing a direct answer, SCoT-Reasoner first decomposes the query into structured reasoning steps that explicitly reference object attributes, spatial relationships, and contextual cues within the 3D

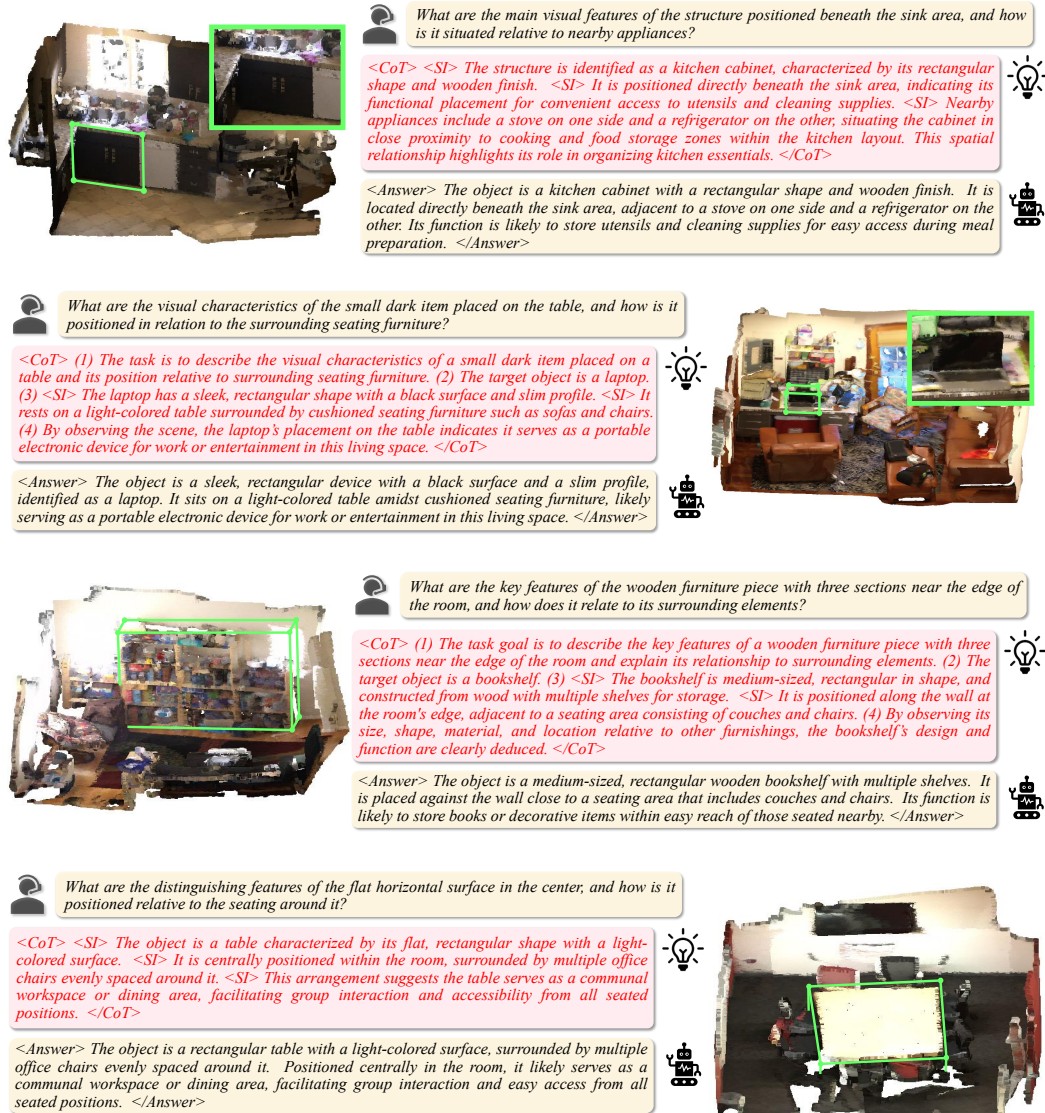

Figure 11: Additional qualitative results of SCoT-Reasoner on object analysis task.

scene. For instance, when asked about the features of a wooden furniture piece, the CoT guides the model to progressively identify its shape, size, material, and spatial positioning relative to surrounding objects, before concluding its functional role. This stepwise reasoning not only ensures that the answer is grounded in observable evidence but also makes the logic of the response transparent in the object analysis task.

As illustrated in Fig. 12, the additional qualitative results of scene analysis demonstrate that CoT reasoning enables 3D LLMs to perform fine-grained scene understanding and logical analysis. Given a raw 3D environment, SCoT-Reasoner first extracts structured information by identifying key spatial elements such as furniture composition, accessibility pathways, and object arrangements in the scene. Then, CoT guides the reasoning process by decomposing the question into interpretable steps, aligning the extracted spatial cues with the semantic intent of the query. For instance, when asked about improving workspace functionality, SCoT-Reasoner not only detects the suboptimal desk placement and lighting but also infers practical modifications, such as repositioning furniture and optimizing light exposure. Similarly, when evaluating whether a seating layout promotes social interaction, the model interprets both the geometry of furniture and the affordances of open space, and then articulates a balanced judgment regarding comfort and accessibility.

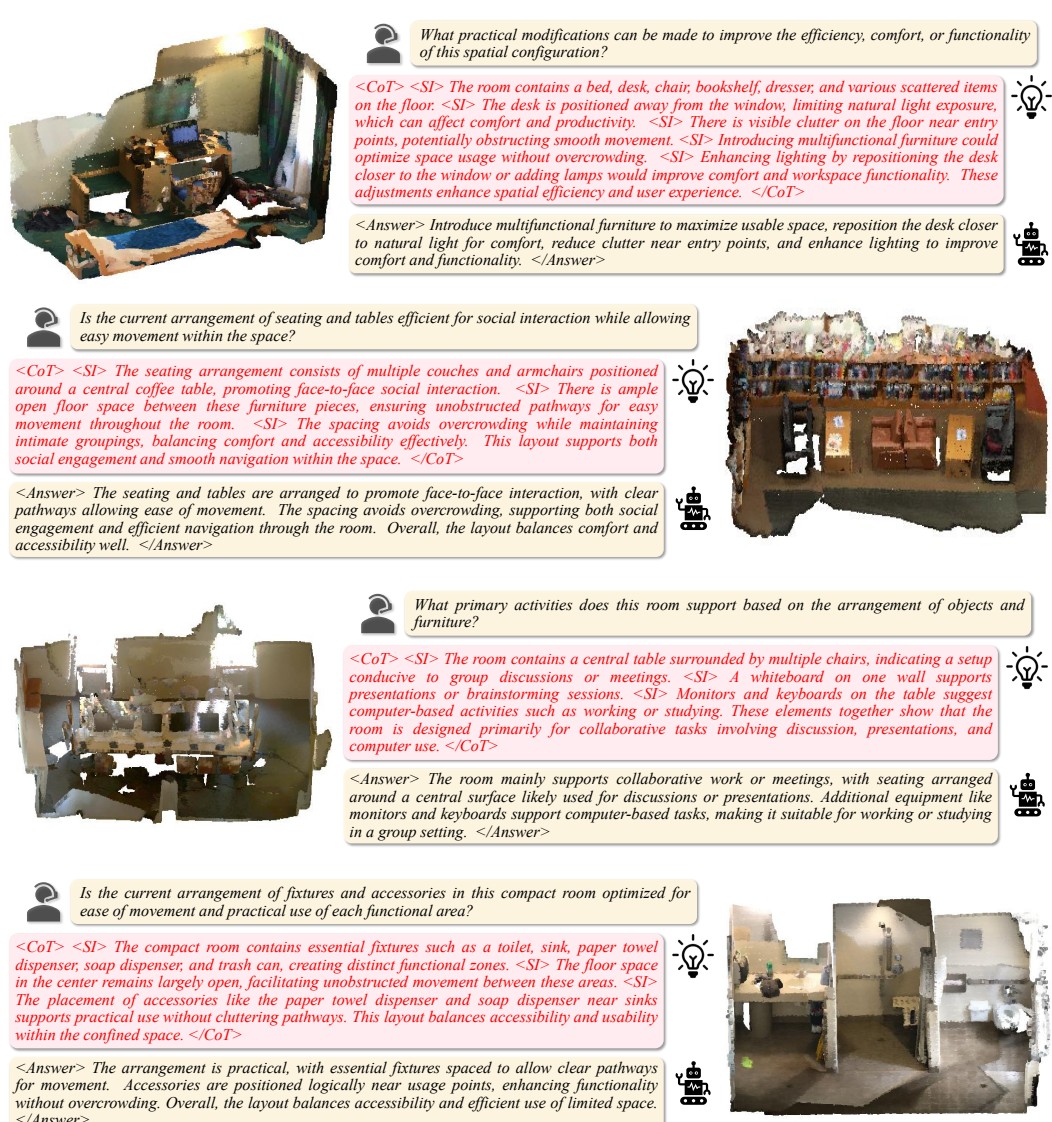

*What practical modifications can be made to improve the efficiency, comfort, or functionality of this spatial configuration?*

*<CoT> <SI> The room contains a bed, desk, chair, bookshelf, dresser, and various scattered items on the floor. <SI> The desk is positioned away from the window, limiting natural light exposure, which can affect comfort and productivity. <SI> There is visible clutter on the floor near entry points, potentially obstructing smooth movement. <SI> Introducing multifunctional furniture could optimize space usage without overcrowding. <SI> Enhancing lighting by repositioning the desk closer to the window or adding lamps would improve comfort and workspace functionality. These adjustments enhance spatial efficiency and user experience. </CoT>*

*<Answer> Introduce multifunctional furniture to maximize usable space, reposition the desk closer to natural light for comfort, reduce clutter near entry points, and enhance lighting to improve comfort and functionality. </Answer>*

*Is the current arrangement of seating and tables efficient for social interaction while allowing easy movement within the space?*

*<CoT> <SI> The seating arrangement consists of multiple couches and armchairs positioned around a central coffee table, promoting face-to-face social interaction. <SI> There is ample open floor space between these furniture pieces, ensuring unobstructed pathways for easy movement throughout the room. <SI> The spacing avoids overcrowding while maintaining intimate groupings, balancing comfort and accessibility effectively. This layout supports both social engagement and smooth navigation within the space. </CoT>*

*<Answer> The seating and tables are arranged to promote face-to-face interaction, with clear pathways allowing ease of movement. The spacing avoids overcrowding, supporting both social engagement and efficient navigation through the room. Overall, the layout balances comfort and accessibility well. </Answer>*

*What primary activities does this room support based on the arrangement of objects and furniture?*

*<CoT> <SI> The room contains a central table surrounded by multiple chairs, indicating a setup conducive to group discussions or meetings. <SI> A whiteboard on one wall supports presentations or brainstorming sessions. <SI> Monitors and keyboards on the table suggest computer-based activities such as working or studying. These elements together show that the room is designed primarily for collaborative tasks involving discussion, presentations, and computer use. </CoT>*

*<Answer> The room mainly supports collaborative work or meetings, with seating arranged around a central surface likely used for discussions or presentations. Additional equipment like monitors and keyboards support computer-based tasks, making it suitable for working or studying in a group setting. </Answer>*

*Is the current arrangement of fixtures and accessories in this compact room optimized for ease of movement and practical use of each functional area?*

*<CoT> <SI> The compact room contains essential fixtures such as a toilet, sink, paper towel dispenser, soap dispenser, and trash can, creating distinct functional zones. <SI> The floor space in the center remains largely open, facilitating unobstructed movement between these areas. <SI> The placement of accessories like the paper towel dispenser and soap dispenser near sinks supports practical use without cluttering pathways. This layout balances accessibility and usability within the confined space. </CoT>*

*<Answer> The arrangement is practical, with essential fixtures spaced to allow clear pathways for movement. Accessories are positioned logically near usage points, enhancing functionality without overcrowding. Overall, the layout balances accessibility and efficient use of limited space. </Answer>*

Figure 12: Additional qualitative results of SCoT-Reasoner on scene analysis task.

As illustrated in Fig. 13, the additional qualitative results of spatial planning highlight the pivotal role of CoT reasoning in enabling 3D LLMs to bridge perception and planning in complex environments. SCoT-Reasoner explicitly verbalizes the current situational state, such as the agent's relative position, available pathways, and the spatial organization of surrounding objects. This step-by-step reasoning allows the system to transform low-level geometry into structured semantic cues, ensuring that task-relevant details (e.g., seating layout, desk placement, or container accessibility) are accurately extracted. Building on this structured scene representation, CoT then decomposes user queries into interpretable subproblems, aligning semantic intent with environmental evidence. For instance, when tasked with organizing a group meeting, the reasoning chain highlights seat arrangement, clutter removal, and lighting adjustments, collectively optimizing comfort and interaction. Similarly, when sorting recyclable items, SCoT-Reasoner not only localizes the recycling bin but also justifies the choice through its visual features and unobstructed access path. By articulating these intermediate steps, CoT ensures that the final answers regarding action instructions are not only correct but also interpretable, trustworthy, and grounded in 3D evidence, thereby enhancing both reasoning transparency and task reliability.

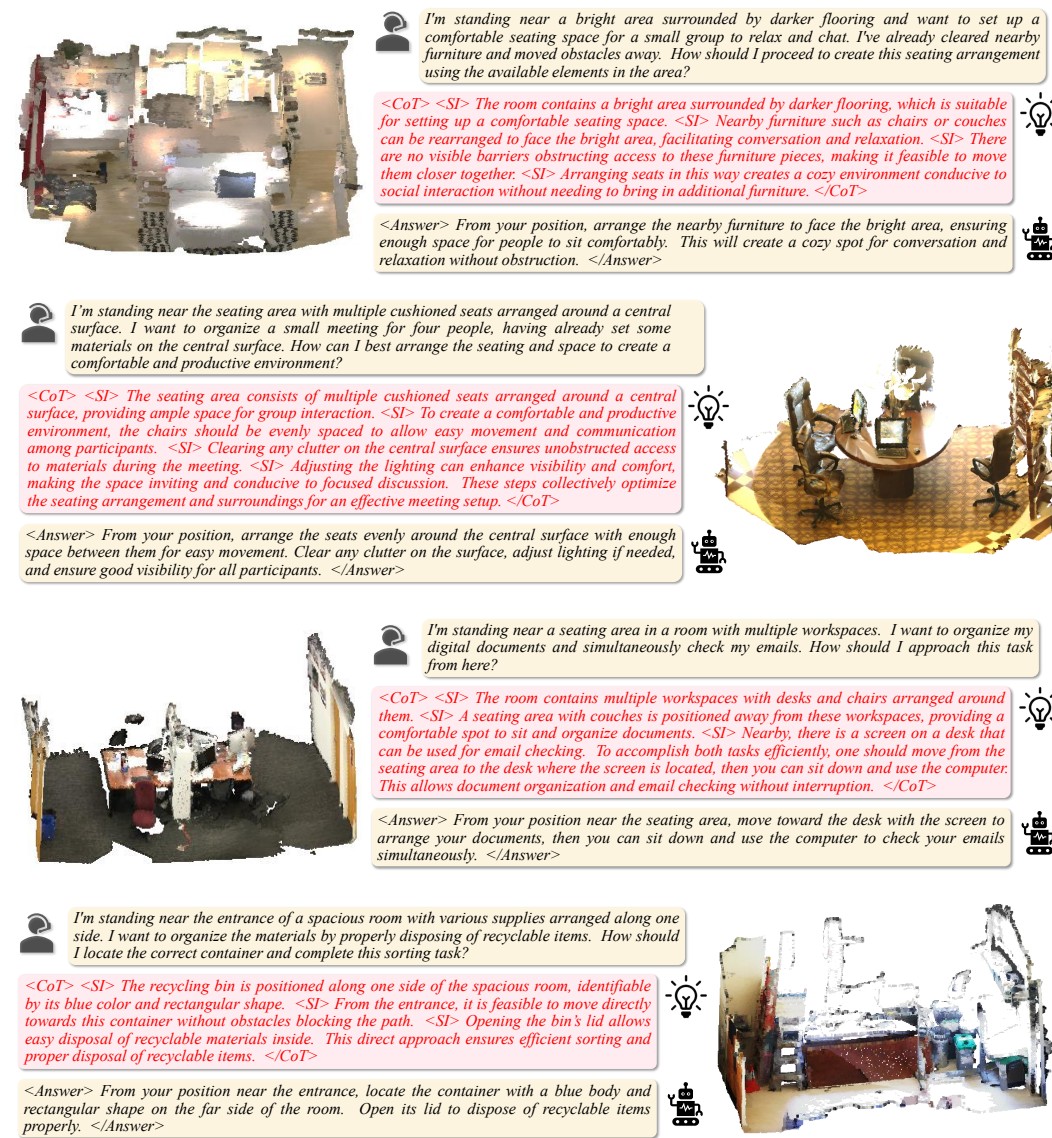

Figure 13: Additional qualitative results of SCoT-Reasoner on un-situated planning and situated planning tasks.

## A.8 BASELINE INTRODUCTION

We re-train the baseline methods (including 3D VG Transformer, 3D-LLM, Chat 3D, Chat 3D V2, LL3DA, Scene-LLM, Chat Scene, and Video 3D LLM) using the same settings with our SCoT-Reasoner for fair comparison in spatial perception, analysis and planning tasks, respectively.

**3D VG Transformer.** 3D VG Transformer is a transformer-based method specifically designed for 3D visual grounding, leveraging diverse relations to facilitate cross-modality proposal disambiguation.

**3D-LLM.** 3D-LLM is the first LLM-based model for 3D scene understanding, which takes 3D point clouds and their dense features derived from multi-view images as input and perform a diverse set of 3D-related tasks.

**Chat 3D.** Chat-3D integrates the perceptual strengths of pre-trained 3D representations with the conversational capabilities of large language models, establishing the first universal dialogue system for 3D scenes.

**Chat 3D V2.** Chat 3D V2 incorporates 3D object representations into LLMs and assigns attribute-aware token and relation-aware token for each object to capture the object's attributes and spatial relationships with surrounding objects in the 3D scene.

**LL3DA.** LL3DA is a large language–based 3D assistant designed to take point clouds as direct input and generate responses to both textual instructions and visual interactions. By incorporating visual interactions, LL3DA effectively understands human engagement within 3D environments, thereby resolving ambiguities that may arise from plain text alone.

**Scene-LLM.** Scene-LLM is a 3D vision–language model designed to enhance the reasoning capabilities of embodied agents. It employs a hybrid 3D visual feature representation that captures dense spatial information while enabling dynamic scene state updates.

**Chat Scene.** Chat Scene improves Chat 3D V2 and receives multimodal representation input, it models the scene embeddings as a sequence of explicit object-level embeddings derived from semantic-rich 2D and 3D representations, performing well in object-level and scene-level tasks.

**Video 3D LLM.** Video 3D LLM treats 3D scenes as dynamic videos, then incorporates 3D position encoding into video representations to aligns video representations with real-world spatial contexts, thereby performing 3D-VL tasks based on 2D Video LLM.

## A.9    LIMITATIONS

While SCoT raises strong reasoning and planning capabilities of 3D-LLMs, several limitations remain. First, the current framework is primarily validated on indoor datasets, and its performance in real-world, open-ended 3D environments requires further investigation. Second, although the CoT reasoning improves interpretability, it may introduce longer inference time and occasional inconsistencies across reasoning steps. Our future work will extend SCoT toward more diverse and dynamic city-scale scenarios.

## A.10    THE USE OF LARGE LANGUAGE MODELS

We clarify the role of Large Language Models in the preparation of this work as follows:

• *Manuscript Writing*: LLMs were only employed for improving grammar and expression during manuscript writing.

• *Research Design and Experiments*: LLMs were not involved in idea conception, methodological design, or experimental implementation.

• *Dataset Generation and Evaluation*: LLMs were used solely as auxiliary tools for dataset generation and evaluation metrics.

