# OpenReview forum: "SCoT: Teaching 3D-LLMs to Think Spatially with Million-scale CoT Annotations"
_ICLR.cc/2026/Conference — ICLR 2026 Poster_

### Official Review · Reviewer_VyFW · 2025-10-30

**Soundness:** 3
**Presentation:** 3
**Contribution:** 3
**Rating:** 6
**Confidence:** 4

**Summary:**

This paper introduces SCOT, a new million-scale dataset designed to teach 3D-LLMs to reason spatially. To address the limitations of existing datasets that lack explicit reasoning steps, SCOT provides structured Chain-of-Thought annotations across a three-level taxonomy of tasks: Spatial Perception, Spatial Analysis, and Spatial Planning. A key contribution is the use of scene-grounded annotations, which force the model to base its reasoning on verifiable 3D evidence, thus improving the transparency and accuracy of complex analysis and planning tasks. Experiments show that this method significantly enhances the model ability to perform complex spatial reasoning, while also demonstrating that overuse of CoT for simple perception can be detrimental.

**Strengths:**

This paper introduces SCOT, a new million-scale dataset designed to teach 3D-LLMs to reason spatially. To address the limitations of existing datasets that lack explicit reasoning steps, SCOT provides structured Chain-of-Thought annotations across a three-level taxonomy of tasks: Spatial Perception, Spatial Analysis, and Spatial Planning. A key contribution is the use of scene-grounded annotations, which force the model to base its reasoning on verifiable 3D evidence, thus improving the transparency and accuracy of complex analysis and planning tasks. Experiments show that this method significantly enhances the model ability to perform complex spatial reasoning, while also demonstrating that overuse of CoT for simple perception can be detrimental.

**Weaknesses:**

The scene-grounded reasoning lacks strong evidence, which relies on LLM judge evaluations rather than direct verification against ground-truth data, which may lead some problems on robustness and objectivity.

**Questions:**

What is the impact of generating detailed CoT reasoning on inference efficiency, and how does this affect the suitability for real-time applications such as robotics?

---

> ### Author Response · Authors · 2025-11-26
> **Responses to Reviewer VyFW**
>
> **We sincerely appreciate the reviewer's insightful comments and time dedicated to evaluating our work. Our responses are listed below.**
>
> **Q1:** Reliability of spatial reasoning in LLM-based judgement.
>
> **A1:** Thank you for raising this concern. We agree that LLM-based judgement on scene-grounded accuracy may be unreliable. Therefore, we explicitly incorporate reliability controls in both the CoT construction and evaluation stages.
>
> ***a) Scene-grounded verification during data construction.***
> The CoT traces undergo scene information (\<SI\>) checks, cross-model validation (GPT-4.1, Qwen, DeepSeek), and manual inspection, achieving an acceptance rate of 90\%. These verification not only guarantees data reliability but also better guides the model to ground its reasoning on explicit 3D scene information.
>
> ***b) Evaluation with bias mitigation and traditional metrics.***
> At test time, we average judgments across multiple heterogeneous LLM evaluators to reduce individual-model bias. We additionally report traditional metrics (e.g., grounding accuracy) to provide LLM-independent validation. Qualitative examples further show that the generated reasoning remains consistent with the underlying scene evidence.
>
> **Q2:** Impact of CoT generation on inference efficiency and real-time applicability.
>
> **A2:** Thank you for the insightful question. We agree that generating detailed CoT reasoning introduces additional computational overhead. Below we quantify the impact and discuss implications for real-time robotics.
>
> ***a) CoT increases inference latency by 2.0×–3.2×.***
> As shown in **Table 9** of revised paper **(Sec.A.6)**, across all tasks, CoT increases inference time by 2.0×–3.2× (e.g., 5.08→13.30s for scene analysis; 6.15→14.27s for un-situated planning). These tasks require multi-step spatial reasoning and CoT analysis to substantially improve the accuracy. However, under our hardware setup (single NVIDIA A100), this level of latency is not yet suitable for strict real-time robotic deployment.
>
> ***b) Potential solutions.***
> Real-time applicability can be further improved through model compression (e.g., quantization and lightweight distillation), hardware-aware optimization, and task-adaptive reasoning. For instance, real-time perception and reactive behaviors can operate in a no-CoT mode, while complex semantic analysis or long-horizon planning selectively enables CoT reasoning. We leave the systematic integration of these techniques as future work to enhance real-time performance.

---

> > ### Comment · Reviewer_VyFW · 2025-11-27
> >
> > Thanks for answering my questions and I will keep my rating unchanged

---

### Official Review · Reviewer_A8Pi · 2025-10-30

**Soundness:** 2
**Presentation:** 3
**Contribution:** 2
**Rating:** 4
**Confidence:** 4

**Summary:**

This work introduces a million-scale chain-of-thought dataset aimed at improving the spatial reasoning capabilities of existing 3D LLMs, along with a baseline model, SCoT-Reasoner. Experiments show that models fine-tuned on the SCoT dataset exhibit performance improvements across various 3D visual question answering benchmarks. Additionally, these fine-tuned models exhibit reasoning processes that are transparent, faithful to the scene, and inherently more trustworthy.

**Strengths:**

1. The proposed SCoT dataset covers a diverse set of spatial tasks and is large in scale. The paper provides sufficient details to enable the research community to reproduce the data generation pipeline.
2. The paper is well-presented and easy to follow.

**Weaknesses:**

1. SCoT is based on only 800 scenes from ScanNet, which is a relatively small scale. While the number of CoT examples is large, I’m concerned whether such a limited range of scene samples can genuinely enhance models' spatial reasoning ability on unseen real-world 3D environments.
2. In Section 3, the author describes the SCoT format as “Query–CoT–Answer.” However, the spatial perception data seems to include only QA pairs without explicit reasoning. Clarification is needed on how these are treated as CoT examples.
3. Most experiments are conducted on the SCoT test set. It would strengthen the paper if the authors included additional evaluations on out-of-domain datasets such as MSR3D [1] and Hypo3D [2] to assess generalization.
4. The authors are encouraged to provide more discussion, ideally supported by results, demonstrating that fine-tuning on SCoT offers greater benefits for spatial reasoning compared to fine-tuning on previous 3D SCoT datasets.

[1] Linghu, Xiongkun, et al. "Multi-modal situated reasoning in 3d scenes." NeurIPS 2024.

[2] Mao, Ye, et al. "Hypo3D: Exploring Hypothetical Reasoning in 3D." ICML 2025.

**Questions:**

1. Throughout the paper, the three main tasks are introduced in the order of SCoT-Perception, SCoT-Analysis, and SCoT-Planning. Why is a different order used in the results section (Section 5.2)? Maintaining consistency would improve readability.

2. It is unclear which model is evaluated in Table 5. The authors should clarify the model configuration or variant used for these results.

3. SCoT-Reasoner appears to be a key model in the evaluation, yet most of its technical details are relegated to the appendix. I recommend moving more of these details into the main Method section to help readers better understand the approach.

**Details Of Ethics Concerns:**

None.

---

> ### Author Response · Authors · 2025-11-26
> **Responses to Reviewer A8Pi (Part 1)**
>
> **We sincerely appreciate the reviewer's encouraging comments and time dedicated to evaluating our work. We make the responses to each question below.**
>
> **Q1:** Verify generalization to unseen 3D environments.
>
> ## Table: Evaluation results on the MSQA dataset under ScanNet and ARKitScenes environments. †: zero-shot test.
> ### **MSQA–ScanNet**
> | Method | Counting | Existence | Attributes | Spatial | Navigation | Others | Overall |
> |--------|----------|-----------|-----------|---------|-----------|--------|--------|
> | **(a) LEO** | 32.5 | 88.5 | 58.7 | 44.2 | 39.6 | 81.4 | 54.8 |
> | **(b) MSR3D** | 32.3 | **93.1** | 50.0 | 46.5 | 54.1 | 75.6 | 54.2 |
> | **(c) LEO-VL** | 39.3 | 92.7 | 56.9 | **59.3** | **59.7** | 82.8 | **61.7** |
> | **(d) GPT-4o†** | 32.3 | 79.3 | **79.0** | 37.0 | 31.7 | **91.6** | 52.3 |
> | **(e) Chat Scene† (3D-R1)** | 35.3 | 41.9 | 34.4 | 45.1 | 25.2 | 86.2 | 43.1 |
> | **(f) Chat Scene† (SCoT)** | 35.5 | 64.8 | 44.5 | 42.7 | 39.8 | 84.8 | 47.6 |
> | **(g) SCoT-Reasoner† (SCoT)** | **41.5** | 63.5 | 54.8 | 48.9 | 43.0 | 91.2 | 54.4 |
> ### **MSQA–ARKitScenes**
> | Method | Counting | Existence | Attributes | Spatial | Navigation | Others | Overall |
> |--------|----------|-----------|-----------|---------|-----------|--------|--------|
> | **(h) GPT-4o†** | 37.5 | **55.2** | 48.1 | 37.7 | 21.0 | 60.7 | 41.0 |
> | **(i) Qwen-VL†** | **43.2** | 46.0 | 44.5 | 25.3 | 26.9 | **70.1** | 39.7 |
> | **(j) SCoT-Reasoner† (SCoT)** | 31.2 | 47.5 | **52.6** | **40.5** | **29.5** | 65.9 | **41.2** |
>
> **A1:** Thank you for your insightful comments. We agree that our goal is to learn scene-grounded reasoning patterns, not scene memorization. To validate this, we conduct out-of-domain generalization experiments using the MSQA benchmark from MSR3D as shown in above table **(Table 6 of the revised paper)**, which includes two distributions: ***MSQA–ScanNet*** with the same scenes as SCoT but different question formulations, agent-centric instructions, and linguistic structures; and ***MSQA–ARKitScenes*** with entirely different environments, camera properties, reconstruction quality, and spatial layouts from SCoT.
>
> ***a) Generalization to Out-of-Domain ARKitScenes.***
> Although trained only on ScanNet, SCoT-Reasoner transfers well to ARKitScenes, a distinct domain. This confirms that SCoT teaches transferable reasoning mechanisms rather than overfitting to ScanNet. As shown in **cases(h)(i)(j)** of above table, SCoT-trained 7B model achieves an overall score of 41.2, exceeding much larger LVLMs (GPT-4o, Qwen-VL) in zero-shot. Especially, our model achieves large gains (+3.3) on reasoning-intensive categories (Attributes, Spatial, Navigation), where scene-grounded inference is essential. Also, qualitative examples in **Figure 10** of the revised paper show consistent, multi-step reasoning grounded in novel ARKitScenes geometry, despite domain shift.
>
> ***b) Generalization to New Question Formulations.***
> As shown in **cases(a)(b)(d)(g)** of above table, on the same underlying ScanNet but with markedly different question styles, SCoT-Reasoner still generalizes well. SCoT-trained model achieves overall score of 54.4, matching matching MSQA-supervised in-domain models (LEO, MSR3D) and largely outperforming outperforms GPT-4o by +2.1 in zero-shot test. Thus, the change in phrasing, state definitions, and agent-centric framing does not harm performance, showing no reliance on training linguistic templates.
>
> We have added the above results and analysis in **Sec.5.4** of the revised paper.
>
> **Q2:** Clarify the CoT annotations in perception tasks.
>
> **A2:** We appreciate the reviewer for pointing this out. In our annotation, we also generated “Query-CoT-Answer” for perception tasks using the same pipeline. However, we found that adding CoT to simple perception tasks introduces unnecessary reasoning and causes hallucinations, leading to a clear accuracy drop (–4.9\% across multiple benchmarks in **Table 2** of the revised paper). Therefore, we keep only the answer during training while still releasing the perception CoT for verification. We have added clarification in **Sec.3.1** and supporting results in **Sec.5.2**.
>
> **Q3:** Compare SCoT with prior 3D CoT datasets in enhancing spatial reasoning.
>
> Thank you for this suggestion. As presented in **Sec.5.4**, we fine-tune a same 3D-LLM **Chat Scene** separately on 3D-R1 (Scene-30k) and SCoT under strictly identical training configurations. Then, we evaluate both models in a zero-shot setting on the out-of-domain MSQA benchmark.
>
> Above table (Table 6 of the revised paper) shows that **SCoT-trained model (f)** outperforms **3D-R1-trained model (e)**, with particularly large improvements in reasoning-sensitive categories: Existence (+22.9), Attributes (+10.1), and Navigation (+14.6). These categories require multi-step planning, relation tracking, and grounded inference. This confirms that SCoT provides stronger and more generalizable spatial reasoning supervision than prior 3D CoT datasets.

---

> ### Author Response · Authors · 2025-11-26
> **Responses to Reviewer A8Pi (Part 2)**
>
> **Q4:** Improving writing order consistency, clarifying model usage and method details.
>
> **A4:** Thank you for the constructive suggestions. We have revised the manuscript to improve clarity, consistency, and completeness as follows.
>
> ***a) Task order consistency.***
> We unified the presentation order of the three tasks to **“SCoT-Perception → SCoT-Analysis → SCoT-Planning”** across both **Sec.3.1** and **Sec.5.2**. Due to space limits, detailed quantitative perception results without CoT requirements are moved to the Appendix **(Sec.A.5)**, and this organization is now explicitly stated in the main text.
>
> ***b) Model used in Table 5.***
> **Table 5 (Table 2 in the revised version)** reports the performance of the proposed ***SCoT-Reasoner***. We clarified the exact model configuration in both the table caption and the method description.
>
> ***c) Placement of SCoT-Reasoner details.***
> Due to space limitations, the full architecture and implementation details of SCoT-Reasoner are provided in the Appendix **(Sec.A.2)** to reveal the details and effectiveness of SCoT. In the revised **Sec.4**, we added a discussion of SCoT-Reasoner: most prior methods focus on a single spatial modality (e.g., point clouds or images). To more comprehensively evaluate SCoT, we design SCoT-Reasoner, a unified framework supporting multimodal inputs.

---

### Official Review · Reviewer_qGaY · 2025-10-31

**Soundness:** 2
**Presentation:** 2
**Contribution:** 2
**Rating:** 2
**Confidence:** 4

**Summary:**

This paper propose SCoT, a large-scale CoT datasets for spatial reasoning. It spanning three levels including spatial perception, spatial analysis and spatial planning. SCoT annotates intermediate reasoning grounded in scene cues. It introduces<SI> token let CoT grounded in scenes context, and reduce hallucinations. Results shows model trained on their proposed datasets benefits complex analysis and planning.

**Strengths:**

1. This paper constructed a valuable large-scale datasets which including diverse scenes with their classified three-tier taxonomy of 3D task.

2. This paper propose a detailed data construction pipeline which provides empirical insights for spatial reasoning data construction.

**Weaknesses:**

1. The novelty is not very clear in this paper, could the author emphasize the key differences between this work and previous spatial reasoning CoT studies, as listed in Table 1 such as 3D-R1 and SpaceR-151k, they are also including multi tasks and reasoning, so what's the biggest advantage of your dataset compared with them? Have you compared with these datasets in controlled settings?

2. The base model used to train is too weak to verify the usefulness of the proposed method. I don't understand why using Vicuna-7B as the pretrained model even in 2025 today.

3. I doubt the validity of the evaluation. If I understand correctly, part of the evaluation data are self-constructed by yourself? (In the table 2, and table 3.) The public evaluation data only appear in Table 7, but over half of results in Table 7 your proposed method can't beat other methods. I think it's extremely unfair to compare with other methods if using your self-built data in the main results, considering we don't know whether the performance gain comes from in distribution benefits. Are training and test scenes completely disjoint?

4. Using same models (GPT-4.1, DeepSeek, Qwen) generating training data as evaluator, which will further introduce bias in evaluation.

**Questions:**

Refer to weakness.

---

> ### Author Response · Authors · 2025-11-26
> **Responses to Reviewer qGaY (Part 1)**
>
> **We sincerely appreciate the reviewer’s feedback and the time dedicated to evaluating our work. We address the concerns as follows.**
>
> **Q1:** Compare SCoT with concurrent reasoning datasets.
>
> **A1:** Thank you for your comments. We clarify that **3D-R1 (ICLR 2026 submission)** is concurrent, and **SpaceR-151k is an ArXiv preprint**. While not required (https://iclr.cc/Conferences/2026/ReviewerGuide), we made the following comparisons to highlight SCoT’s advantages.
>
> ***a) Principled task taxonomy:*** Unlike other CoT datasets, SCoT’s annotations are designed according to the task-specific needs of CoT. We introduce a structured “perception-analysis-planning” taxonomy. On the one hand, we provide answer-only supervision for perception because excessive CoT can cause hallucinations. On the other hand, we reveal that scene-grounded CoTs are necessary for complex analysis and planning. As shown in **Table 2 (Sec.5.2)** of the revised paper, without this design, indiscriminate CoT on perception tasks leads to a 4.9\% accuracy drop.
>
> ***b) Million-level volume with rigorous quality control:*** As shown in **Table 1 (Sec.1)** of the revised paper, SCoT contains 1.1M samples, far exceeding 3D-R1 (Scene-30k) (30k), SpaceR-151k (151k), and Spatial-MLLM-120k (120k). Each reasoning step is explicitly grounded via \<SI\>, and samples failing this constraint are filtered via multi-LLM cross-checks.
>
> ***c) Quantitative comparison with 3D-R1:*** In **Sec.5.4**, we train a 3D-LLM Chat Scene on 3D-R1 and SCoT under identical settings, and conduct zero-shot evaluation on the out-of-domain MSQA benchmark to fairly and accurately assess reasoning ability. As shown in **Table 6 (e) and (f)** of the revised paper, the results show that the SCoT-trained model outperforms the 3D-R1-trained model, with particularly large gains in reasoning-sensitive categories such as Navigation (+14.6).
>
> We have marked the publication status for these works in **Table 1 (Sec.1)**, and added the results and discussions in **Sec.5.4**.
>
> **Q2:** Reasons for choosing a 7B LLM (Vicuna-7B).
>
> **A2:** Thank you for your comments. We clarify that Vicuna-7B-v1.5 was chosen for fairness and controlled evaluation of SCoT’s contribution (as also recognized by reviewers sqpn, A8Pi and VyFW), not only for state-of-the-art performances.
>
> Representative 3D-LLM baselines, ***even in 2025 today[1-7]***, use Vicuna-7B as backbone due to hardware constraints and the trade-off between expressiveness and trainability. Using Vicuna-7B avoids confounding effects from heterogeneous models or parameter scales, isolating the dataset’s impact. Results show that SCoT consistently enhances explainability, faithfulness, trustworthiness, and implicit detection across several representative models **(Tables 3-5 in Sec. 5.2)**, regardless the backbone choise or architecture designs. Consequently, using Vicuna-7B in our SCoT-Reasoner preserves architectural consistency across all baselines and enables a fair, controlled comparison.
>
> Training larger models is currently beyond our computational budget. We leave this as future work, but we expect SCoT’s reasoning annotation scheme and methodology to provide valuable insights for scaling to larger models.
>
> *[1] Yu H, Li W, Wang S, et al. “Inst3d-lmm: Instance-aware 3d scene understanding with multi-modal instruction tuning” CVPR 2025.*
>
> *[2] Kang W, Huang H, Shang Y, et al. “Robin3d: Improving 3d large language model via robust instruction tuning” ICCV 2025.*
>
> *[3] Deng J, He T, Jiang L, et al. “3D-LLaVA: Towards generalist 3d lmms with omni superpoint transformer” CVPR 2025.*
>
> *[4] Ahmed M, Fei J, Ding J, et al. “Kestrel: 3D Multimodal LLM for Part-Aware Grounded Description” ICCV 2025.*
>
> *[5] Wang Y, Chen Y, Qi Z, et al. “Mamba-3VL: Taming State Space Model for 3D Vision Language Learning” ICCV 2025.*
>
> *[6] Zhu H, Kong Q, Xu K, et al. “Grounding 3D Object Affordance with Language Instructions, Visual Observations and Interactions” CVPR 2025.*
>
> *[7] Choi C, Shin Y, Han G, et al. “B4DL: A Benchmark for 4D LiDAR LLM in Spatio-Temporal Understanding” ACM MM 2025.*

---

> ### Author Response · Authors · 2025-11-26
> **Responses to Reviewer qGaY (Part 2)**
>
> **Q3:** Evaluation validity and dataset splits.
>
> **A3:** Thank you for your comments. We clarify several points regarding dataset construction and evaluation validity.
>
> ***a) Strict scene disjointness.***
> SCoT is built following the same ScanNet split as ScanRefer, SQA3D, ScanQA, and Scan2Cap. We use 562 training scenes and 239 completely disjoint test scenes. All tasks (spatial perception, analysis, and planning) are generated from these standardized scenes. Thus, no scene-level leakage or in-distribution advantage exists, and we will open-source it.
>
> ***b) Fair comparison under controlled distributions.***
> In the revised paper, tasks in **Tables 3-4** (Tables 2-3 in original paper) share the same scenes and distributions as the scenes used in **Table 8** (Table 7 in original paper). The only difference is task difficulty. This ensures that analysis and planning performance gains arise from improved reasoning rather than distributional bias.
>
> ***c) Table 8 (Table 7 in original paper) focuses on perception, not reasoning.***
> **Table 8** assesses simple spatial perception, where answers rely on direct scene lookup rather than the multi-step reasoning targeted by SCoT. As stated in **Sec.4** and **Sec.A.2** of the revised paper, SCoT-Reasoner is designed to support multi-modal inputs (e.g., text, point cloud, and video), together with other baselines, to provide a comprehensive, architecture-agnostic evaluation of SCoT in **Tables 3-4** (Tables 2-3 of the original paper), not only for SoTA accuracy. However, even supervised with simple perception tasks (without CoT), SCoT-Reasoner still outperforms other methods in nearly half of the results benefited by multi-modal input designs, particularly in 3D grounding.
>
> ## Table: Evaluation results on the MSQA test set under ScanNet and ARKitScenes environments. †: zero-shot test.
> ### **MSQA–ScanNet**
> | Method | Counting | Existence | Attributes | Spatial | Navigation | Others | Overall |
> |--------|----------|-----------|------------|---------|------------|--------|---------|
> | **(a) LEO** | 32.5 | 88.5 | 58.7 | 44.2 | 39.6 | 81.4 | 54.8 |
> | **(b) MSR3D** | 32.3 | **93.1** | 50.0 | 46.5 | 54.1 | 75.6 | 54.2 |
> | **(c) LEO-VL** | 39.3 | 92.7 | 56.9 | **59.3** | **59.7** | 82.8 | **61.7** |
> | **(d) GPT-4o†** | 32.3 | 79.3 | **79.0** | 37.0 | 31.7 | **91.6** | 52.3 |
> | **(e) Chat Scene† (3D-R1)** | 35.3 | 41.9 | 34.4 | 45.1 | 25.2 | 86.2 | 43.1 |
> | **(f) Chat Scene† (SCoT)** | 35.5 | 64.8 | 44.5 | 42.7 | 39.8 | 84.8 | 47.6 |
> | **(g) SCoT-Reasoner† (SCoT)** | **41.5** | 63.5 | 54.8 | 48.9 | 43.0 | 91.2 | 54.4 |
> ### **MSQA–ARKitScenes**
> | Method | Counting | Existence | Attributes | Spatial | Navigation | Others | Overall |
> |--------|----------|-----------|------------|---------|------------|--------|---------|
> | **(h) GPT-4o†** | 37.5 | **55.2** | 48.1 | 37.7 | 21.0 | 60.7 | 41.0 |
> | **(i) Qwen-VL†** | **43.2** | 46.0 | 44.5 | 25.3 | 26.9 | **70.1** | 39.7 |
> | **(j) SCoT-Reasoner† (SCoT)** | 31.2 | 47.5 | **52.6** | **40.5** | **29.5** | 65.9 | **41.2** |
>
> ***d) Remarkable results when generalized to other benchmarks.***
> As shown in above table **(Table 6 of revised paper)**, our model trained on SCoT achieves strong zero-shot performance on the MSQA benchmark across both cross-scene and cross-linguistic settings, surpassing large-scale multimodal LLMs (GPT-4o and Qwen-VL) and approaching the accuracy of MSQA-trained in-domain models (LEO and MSR3D). These further demonstrate that SCoT provides reliable and generalizable reasoning capabilities.
>
>
> **Q4:** Mitigating potential LLM bias in evaluation.
>
> **A4:** We agree that LLMs may have bias. To mitigate this, we deliberately adopt a more diverse evaluation pipeline than prior works (e.g., **MSR3D[1]**), which rely on a single ChatGPT model for both annotation and testing.
>
> ***a) Diverse version LLMs for annotation and evaluation.***
> For annotation, we use a heterogeneous model set (GPT-4.1-mini, Qwen-VL-plus, DeepSeek-Chat) with manual verification to ensure scene-faithful samples. For evaluation, we intentionally switch to higher-tier models (GPT-4.1, Qwen-VL-Max, DeepSeek-Reason) to prevent bias propagation from annotation.
>
> ***b) Inter-evaluator consistency analysis.***
> As suggested by the reviewer sqpn, we examine consistency across evaluators and find only moderate correlations (0.3–0.7), shown in **Figure 9 (Sec.A.4)**. Averaging their scores effectively reduces LLM-specific bias.
>
> ***c) Unbiased traditional metrics.***
> Finally, our main conclusions are also supported by unbiased traditional metrics (Acc@0.25, Acc@0.50, ROUGE-L, METEOR), offering orthogonal validation.
>
> *[1] Linghu X, Huang J, Niu X, et al. "Multi-modal situated reasoning in 3d scenes." NeurIPS 2024.*

---

### Official Review · Reviewer_sqpn · 2025-11-01

**Soundness:** 3
**Presentation:** 3
**Contribution:** 3
**Rating:** 6
**Confidence:** 4

**Summary:**

This paper presents SCoT, a million-scale Chain-of-Thought (CoT) dataset designed to teach 3D Large Language Models (3D-LLMs) to reason spatially.
SCoT organizes tasks into three tiers — Spatial Perception (“what is there”), Spatial Analysis (“what does it mean”), and Spatial Planning (“what should I do”) — and annotates CoT reasoning only where necessary.
The dataset introduces scene-grounded reasoning with explicit \<SI> tags to ensure factual alignment with 3D context.
Experiments on multiple baselines (Chat3D, ChatScene, Video3D-LLM) and the proposed SCoT-Reasoner show significant improvements in reasoning explainability, faithfulness, and planning accuracy.

**Strengths:**

High-quality dataset: 1.1M diverse samples covering perception, reasoning, and planning with strong annotation rigor and cross-checking.

Innovative design: The three-tier CoT taxonomy (perception–analysis–planning) effectively balances reasoning depth and hallucination control.

Grounded CoT methodology: The \<SI> tag mechanism enforces transparent, scene-based reasoning rather than textual hallucination.

Strong empirical validation: Extensive quantitative and qualitative analyses demonstrate consistent gains in complex reasoning tasks.

Practical impact: Provides a scalable framework for training reliable, interpretable 3D-LLMs relevant to embodied AI and robotics.

Solid writing and clarity: The paper is well-organized, and figures (e.g., Fig. 1 & 3) effectively illustrate the framework and dataset pipeline.

**Weaknesses:**

The use of LLM-based evaluators (ChatGPT, Qwen, DeepSeek) for “Explainability,” “Faithfulness,” and “Trustworthiness” is well-motivated but inherently subjective.
It would strengthen credibility if the authors validated inter-evaluator consistency (e.g., correlation scores between evaluators).

The paper does not evaluate how models trained on SCoT generalize to unseen 3D environments or other datasets (e.g., ARKitScenes or Omni3D).

The paper could provide more detailed ablations: Comparing models trained with different CoT lengths or varying levels of \<SI> grounding.；Analyzing which task types (object vs. scene vs. planning) benefit most from CoT supervision.；Evaluating performance trade-offs when CoT is partially removed during inference.

Although Table 5 highlights hallucination in perceptual CoT, the causes (e.g., linguistic priors vs. visual overfitting) are not deeply analyzed.
This discussion could offer more insight into why CoT sometimes harms visual fidelity.

**Questions:**

See Weaknesses

---

> ### Author Response · Authors · 2025-11-26
> **Responses to Reviewer sqpn (Part 1)**
>
> **We sincerely appreciate the reviewer for their invaluable comments and time dedicated to evaluating our work. We provide our responses to each question below.**
>
> **Q1:** Validate consistency across different LLM evaluators.
>
> **A1:** Thank you for the valuable suggestion. We incorporated an inter-evaluator consistency analysis for ChatGPT-4.1, Qwen, and DeepSeek in **Sec.A.4 (Figure 9)** of the revised paper.
>
> Across both spatial analysis and planning tasks, the pairwise correlations remain in a moderate range (0.3–0.7). This behavior is desirable for two reasons. First, ***the correlations are far from high or degenerate***, indicating that the three heterogeneous evaluators do not share strong coupling or model-specific preferences, and thus no single evaluator biases the overall results. Second, ***the correlations are consistently positive***, showing that the evaluators capture a common assessment signal rather than producing random or unaligned scores.
>
> Together, these findings demonstrate that the proposed metric is neither dominated by a particular model nor driven by noise. To further enhance robustness, we thus report the average score across the three evaluators as the final metric.
>
> **Q2:** Verify generalization to unseen 3D domains.
>
> **A2:** Thank you for raising this important point. To evaluate cross-domain generalization, we conduct zero-shot evulation on the ARKitScene dataset annotated by MSQA in **Sec.5.4**. This setting provides a rigorous out-of-domain test for two reasons:
>
> (1) ***Data discrepancy***: ARKitScenes differs from ScanNet in sensor characteristics, reconstruction fidelity, and scene layout.
>
> (2) ***Q-A discrepancy***: The MSQA question-answer style and task types differ substantially from those in SCoT.
>
> These make both the visual domain and the reasoning tasks unseen during training.
>
> **Table: Evaluation results on MSQA-ARKitScenes. ‡ indicates zero-shot test. SCoT-Reasoner is trained on SCoT.**
> | Method                         | Counting | Existence | Attributes | Spatial | Navigation | Others | Overall |
> |--------------------------------|----------|-----------|------------|---------|------------|--------|---------|
> | GPT-4o‡                       | 37.5     | **55.2**  | 48.1       | 37.7    | 21.0       | 60.7   | 41.0    |
> | Qwen-VL‡                      | **43.2** | 46.0      | 44.5       | 25.3    | 26.9       | **70.1** | 39.7    |
> | SCoT-Reasoner‡ (SCoT)         | 31.2     | 47.5      | **52.6**   | **40.5**| **29.5**   | 65.9   | **41.2** |
>
> Despite these shifts, our model (7B) trained on SCoT achieves superior performance to large MM-LLMs like GPT-4o on spatial analysis and planning tasks in MSQA-ARKitScenes, as shown in above table (**Table 6** in revised paper). This demonstrates that the structured spatial reasoning knowledge acquired from SCoT transfers beyond its source domain.
>
> Moreover, qualitative results **(Figure 10)** show that SCoT-Reasoner consistently produces scene-grounded, logically coherent, and multi-step reasoning before answering. These findings collectively indicate strong generalization to unseen 3D environments.

---

> ### Author Response · Authors · 2025-11-26
> **Responses to Reviewer sqpn (Part 2)**
>
> **Q3:** Provide more ablations and analysis on CoT length, varying levels of \<SI\> grounding, and inference efficiency.
>
> **A3:** Thank you for the insightful suggestions. We have incorporated a comprehensive ablation study in **Table 9** of the revised paper **(Sec.A.6)**.
>
> ***a) Ablating \<SI\> grounding levels (CoT length).*** We categorized, filtered, and ablated different types of \<SI\> annotations. As shown in **Table 9** of the revised paper, “w/o Obj” denotes removing \<SI\> for object-centric information, while “w/o Sce” removes scene-level analysis \<SI\> from the CoT. Results indicate:
>
> - Object-level reasoning is critical for object-centric tasks; removing it consistently reduces performance (e.g., METEOR 16.17→15.34; Explainability 7.04→6.43; Faithfulness 6.15→5.37).
>
> - Scene-level reasoning is essential for holistic scene understanding and planning, with substantial drops when removed (e.g., Scene Analysis METEOR 15.29→14.37; Situated Planning Trustworthiness 7.14→6.55).
>
> ***b) Performance trade-offs when disabling CoT at inference.***
> Full CoT provides the strongest reasoning gains but increases inference latency by 2.0×–3.2× (e.g., Scene Analysis 5.08→13.30s; Un-situated Planning 6.15→14.27s on a single A100). This overhead is acceptable for offline planning but we agree that it is less suitable for time-sensitive robotic applications.
>
> Real-time applicability can be improved via standard acceleration techniques such as quantization, lightweight distillation, and hardware-aware optimization. In addition, task-adaptive reasoning pattern can be integrated into SCoT-Reasoner: real-time perception and reactive behaviors can operate in a no-CoT mode, while complex semantic analysis or long-horizon planning selectively enables CoT reasoning. We leave the systematic integration of these techniques as future work to enhance real-time performance.
>
> **Q4:** Analyze perception hallucinations from CoT.
>
> **A4:** Thank you for the comment. We have expanded the discussion in **Sec.5.2** to analyze two main causes of hallucination in perceptual CoT.
>
> ***a) Linguistic priors overriding scene-grounded evidence.***
> In perception tasks, correct answers depend primarily on visual or geometric cues. CoT can bias 3D-LLMs toward common-sense patterns (e.g., “chairs are usually brown”), causing linguistic priors to override critical scene evidence.
>
> ***b) Unnecessary reasoning steps increasing hallucination risk.***
> Perceptual tasks usually require short, factual answers. CoT introduces multi-step reasoning that is not needed, exposing the model to speculative intermediate states and resulting in hallucinated attributes.

---

### Author Response · Authors · 2025-11-26
**Overall Response from Authors**

We sincerely thank the reviewers for their insightful feedback and constructive suggestions, and appreciate their recognition of our work’s key contributions:

*a) Large-scale, diverse, and rigorously annotated 3D reasoning dataset.*  SCoT provides over 1.1M high-quality samples spanning spatial perception, analysis, and planning tasks (sqpn, qGaY, A8Pi, VyFW).

*b) Structured task taxonomy and principled scene-grounded CoT design.* The three-tier “perception–analysis–planning” taxonomy, together with the \<SI\> tag mechanism, enforces transparent, scene-based reasoning and effectively balances reasoning depth with hallucination control (sqpn, qGaY, A8Pi, VyFW).

*c) Strong empirical reasoning gains across multiple 3D-LLMs.* Models trained on SCoT demonstrate substantial improvements in comprehensive metrics (Explainability, Faithfulness, and Trustworthiness), particularly on complex spatial analysis and planning tasks (sqpn, A8Pi, VyFW).

### **Changes in the main paper include:**

- Publication status of existing 3D-LLM datasets (Table 1) in Sec.1.

- Clarification of the reasons for the CoT absence in spatial perception samples in Sec.3.1.

- 3D-LLM selection and SCoT-Reasoner design in Sec.4.

- Specific models used in spatial perception tasks (Table 2) in Sec.5.2.

- In-depth analysis for the inapplicability of CoT reasoning in spatial perception tasks in Sec.5.2.

- Generalization experiments on the MSR3D (MSQA) dataset in Sec.5.4.

### **Changes in the revised supplementary material include:**

- Inter-evaluator correlation visualizations and analysis among LLM evaluators in Sec.A.4.

- Ablation study (including accuracy and inference efficiency) for different CoT settings and \<SI\> levels in Sec.A.6.

---

### Meta-Review · Area_Chair_bDbC · 2026-01-05

**Summary:**

This paper proposes a new dataset (including both training and evaluation aspects) for performing grounded CoT reasoning using 3D-LLMs.  Reviewers raised several concerns such as whether the benefits of this dataset can be seen in other settings, a potential source of bias in the LLM-based evaluators, and some additional ablations.  The authors provided a robust response, addressing many of these concerns.  Overall- this paper provides a dataset that can help improve understanding of CoT, and is recommended for presentation at the conference.

**Reviewer Concerns:**

For each of the reviewer concerns, the authors provided a reasonable response.  There are some questions as to whether a LLM evaluator can be trusted (the authors discussion on this point indicates it is reasonable, but the extent of its ability is still an open problem), and stronger base models would have been useful, which leaves a risk that the benefits may not generalize to larger models. The benefits on out-of-distribution datasets is good but a little weak, although it can be easily explained away (e.g., differences in pretraining data and model scale).

**Reviewer Scores:**

The outstanding concerns do not represent a strong enough argument to recommend rejection.  As such, I would expect that reviewer scores would likely increase for those initially recommending rejection and either improve or be maintained for the remaining reviewers.

---

### Decision · Program_Chairs · 2026-01-26

Accept (Poster)